# Target alignment in truncated kernel ridge regression

**Arash A. Amini**[1], **Richard Baumgartner**[2], **Dai Feng**[3]

[1]University of California, Los Angeles
[2]Merck & Co., Inc., Rahway, New Jersey, USA
[3]Data and Statistical Sciences, AbbVie Inc.

aaamini@ucla.edu, richard_baumgartner@merck.com,
dai.feng@abbvie.com

## Abstract

Kernel ridge regression (KRR) has recently attracted renewed interest due to its potential for explaining the transient effects, such as double descent, that emerge during neural network training. In this work, we study how the alignment between the target function and the kernel affects the performance of the KRR. We focus on the truncated KRR (TKRR) which utilizes an additional parameter that controls the spectral truncation of the kernel matrix. We show that for polynomial alignment, there is an *over-aligned* regime, in which TKRR can achieve a faster rate than what is achievable by full KRR. The rate of TKRR can improve all the way to the parametric rate, while that of full KRR is capped at a sub-optimal value. This shows that target alignment can be better leveraged by utilizing spectral truncation in kernel methods. We also consider the bandlimited alignment setting and show that the regularization surface of TKRR can exhibit transient effects including multiple descent and non-monotonic behavior. Our results show that there is a strong and quantifiable relation between the shape of the *alignment spectrum* and the generalization performance of kernel methods, both in terms of rates and in finite samples.

## 1   Introduction

Kernel methods have become a time-proven popular mainstay in machine learning [23, 27, 22]. Implicit transformation of a learning problem via a suitable kernel with subsequent development of regularized linear models in the associated reproducing kernel Hilbert space (RKHS) is amenable to applications and allow for investigation of theoretical properties of the kernel methods [12]. Recently, there has been a revived interest and increased research into the kernel methods [14, 17, 7, 12, 8]. These efforts have been motivated by the connections between neural network and kernel learning [14]. In particular, the discovery of transient effects such as double descent that emerge in neural networks training [6, 20] has attracted a lot of attention.

In this paper, we study the effect of target-model alignment on the performance of kernel ridge regression (KRR) estimators. KRR is a well-known nonparametric regression estimator with desirable theoretical performance guarantees. We focus on a generalization of the KRR, which we refer to as the Truncated Kernel Ridge Regression (TKRR), that introduces an additional parameter, $r$, for truncating the eigenvalues of the underlying kernel matrix. TKRR is often used as an approximation to the full KRR to scale the computations to large sample sizes. TKRR itself is often approximated in practice by various sampling and sketching schemes [30, 33, 18, 19, 24, 1, 32]. One of our main contributions in this paper is that TKRR can, in fact, improve the statistical rate of convergence in cases where the target function is well-aligned with the underlying RKHS. Roughly speaking, target

alignment refers to the decay rate of the coefficients of the target function in an orthonormal basis of the eigenfunctions of the kernel operator.

In addition to improved rates for TKRR, we also show that target alignment generally predicts the generalization performance of the full KRR as well as TKRR. We also explore the regularization surface of the KRR estimators in both the regularization and truncation parameters, and show some surprising behavior like non-monotonicity, multiple descent and phase transition phenomena.

**Our main contributions**

(i) We show that TKRR itself can be viewed as a full KRR estimator in a smaller RKHS embedded in the original RKHS (Propostion 1). This shows that the ad-hoc truncation that is often done for computational reasons can itself be viewed as a proper KRR estimator over a different RKHS.

(ii) We introduce target alignment (TA) scores in Section 4 and derive an exact expression for the Mean Sqaured Error (MSE) of TKRR based on the TA scores and eigenvalues of the kernel matrix (Theorem 1).

(iii) For the case of polynomially-decaying TA scores, we first derive a simplified expression for the MSE (Eqn. (14)) which is within constant factors of the exact expression. We then identify four target alignment regimes: Under-aligned, just-aligned, weakly-aligned and over-aligned. We show that TKRR can achieve a fast rate of convergence in the *over-aligned* regime. In contrast, the rate of the full KRR is capped to a sub-optimal level, that of the best achievable in the weakly-aligned regime (Theorem 2).

(iv) For bandlimited TA scores, we provide more refined finite-sample analysis. In particular, we show that the regularization curve as a function of the truncation level $r$ is in general non-monotonic. We also show that TA spectra that are narrower and shifted towards lower indices correspond to lower generalization errors (Proposition 2).

We provide experiments verifying the multiple-descent and non-monotonic behavior of the regularization curves as well as the improved rate of Theorem 2 (Section 4.2). The MSE in our results, is equivalent to the excess (empirical) generalization error. So all the conclusions, and in particular the rates, are valid for the generalization error. Overall, our results provide strong theoretical and experimental support for the idea that alignment of the target function with the eigenfunctions of the kernel is the key predictor of generalization performance of KRR estimators. It is worth noting that our main theoretical results are non-asymptotic and at times exact.

**Related work.** Kernel ridge regression (KRR) has been widely studied in the literature in the non-parametric statistical framework [29, 26]. Various aspects of the KRR were elucidated. For example, the error rates of the KRR were studied in [9] and more recently benign overfitting [5] and related double descent effects in [13].

The notion of kernel-target alignment was first introduced in [11] for classification. In [8], the generalization error of the KRR was investigated using the tools of statistical physics theory which allows for derivation of the bias-variance decomposition of the generalization error and elucidation of non-monotonic transitional effects reflected by the learning curves. In this work the authors have also demonstrated that the interplay between spectral bias (expressed as kernel eigenvalue decay) and kernel-target alignment or task-model alignment are useful for the characterization of the kernel compatibility with the problem. Building on the [8], the concept of kernel-target alignment and spectral bias was referred to as source and capacity, respectively in [12]. Similarly to [8], the transitions that emerge during a crossover from noiseless to noisy regime corresponding to fast vs slow decay of the learning curve, respectively were studied in [12]. Moreover, in [2] it has been shown that the truncated KRR outperforms full KRR over the unit ball of the RKHS. Another related work is that of [25] where overparametrized KRR is studied. The work [12] is perhaps closest to ours followed by [2] and [25]. We make more detailed comparisons with these after Theorem 2. In particular, we show that the rate obtained in [12] can be improved by using TKRR. In addition, in Appendix D, we provide a more unified view of the settings of [9, 12] which can help clarify how different target alignment assumptions relate to each other. It is worth noting that approximate forms of TKRR via sampling and sketching have been analyzed extensively in the literature [21, 32, 10, 31, 15, 1, 4], but this line of work at best proves that TKRR achieves the same minimax rate achievable with full KRR, over the RKHS ball. What we show in this paper is that TKRR can achieve much faster rates under proper target alignment.

## 2  Kernel Ridge Regression

The general nonparametric regression problem can be stated as

$$y_i = f^*(x_i) + \varepsilon_i, \ i = 1, \ldots, n, \quad \mathbb{E}(\varepsilon) = 0, \ \mathrm{cov}(\varepsilon) = \sigma^2 I_n \tag{1}$$

where $\varepsilon = (\varepsilon_i) \in \mathbb{R}^n$ is a noise vector and $f^* : \mathcal{X} \to \mathbb{R}$ is the function of interest to be approximated from the noisy observations $y = (y_i)$. Here, $\mathcal{X}$ is the space to which the *covariates* $\boldsymbol{x} = (x_i)$ belong. We mainly focus on the fixed design regression where the covariates are assumed to be deterministic. A natural estimator is the kernel ridge regression (KRR), defined as the solution of the following optimization problem:

$$\widehat{f}_{n,\lambda} := \operatorname*{argmin}_{f \in \mathbb{H}} \ \frac{1}{n} \sum_{i=1}^{n} (y_i - f(x_i))^2 + \lambda \|f\|_{\mathbb{H}}^2, \tag{2}$$

where $\lambda > 0$ is a regularization parameter and $\mathbb{H}$ is the underlying RKHS. By the representer theorem [16], this problem can be reduced to a finite-dimensional one:

$$\min_{\omega \in \mathbb{R}^n} \ \frac{1}{n} \|y - \sqrt{n} K\omega\|^2 + \lambda \omega^T K\omega, \quad \text{where} \quad K = \frac{1}{n}\big(\mathbb{K}(x_i, x_j)\big) \in \mathbb{R}^{n \times n} \tag{3}$$

is the (normalized empirical) kernel matrix. Let us define the following operators

$$S_{\boldsymbol{x}}(f) = \frac{1}{\sqrt{n}}(f(x_1), f(x_2), \ldots, f(x_n)), \quad S_{\boldsymbol{x}}^*(\omega) = \frac{1}{\sqrt{n}} \sum_{j=1}^{n} \omega_j \mathbb{K}(\cdot, x_j) \tag{4}$$

for $f \in \mathbb{H}$ and $\omega \in \mathbb{R}^n$. We refer to $S_{\boldsymbol{x}}$ as the *sampling operator*. As the notation suggests, $S_{\boldsymbol{x}}^*$ is the adjoint of $S_{\boldsymbol{x}}$ as an operator from $\mathbb{H}$ to $\mathbb{R}^n$. We have $\widehat{f}_{n,\lambda} = S_{\boldsymbol{x}}^*(\widehat{\omega})$ where $\widehat{\omega}$ is the solution of (3). We note that $S_{\boldsymbol{x}}(S_{\boldsymbol{x}}^*(\omega)) = K\omega$, i.e., $K$ is the matrix representation of $S_{\boldsymbol{x}} S_{\boldsymbol{x}}^* : \mathbb{R}^n \to \mathbb{R}^n$.

**Notation.** We write $\langle f, g \rangle_n = \frac{1}{n} \sum_{i=1}^n f(x_i) g(x_i)$ and $\|f\|_n = \sqrt{\langle f, f \rangle_n}$ for the empirical inner product and norm, respectively. Note that $\|f\|_n = \|S_{\boldsymbol{x}}(f)\|_2$ where $\| \cdot \|_2$ is the vector $\ell_2$ norm. For two functions, $f$ and $g$, we write $f \lesssim g$ if there is constant $C > 0$ such that $f \le Cg$, and we write $f \asymp g$ if $f \lesssim g$ and $g \lesssim f$.

## 3  Truncated Kernel Ridge Regression

As mentioned earlier, spectral truncation of the kernel matrix has been mainly considered as a computational device in approximating the full KRR estimator. Here, we first argue that it can be reformulated as a KRR estimator over a simpler RKHS. To facilitate future developments, we define this new RKHS based on the given training data $\boldsymbol{x} = (x_1, \ldots, x_n)$.

Let $K = \sum_{k=1}^n \mu_k u_k u_k^T$ be the eigendecomposition of the kernel matrix $K$ where $\mu_1 \ge \mu_2 \ge \cdots \ge \mu_n \ge 0$ are the eigenvalues of $K$ and $\{u_k\} \subset \mathbb{R}^n$ are the corresponding eigenvectors. We assume for simplicity that $\mu_n > 0$, that is:

**Assumption 1.** *The kernel matrix $K = K(\boldsymbol{x})$ is invertible.*

This is a very mild assumption for any infinite-dimensional $\mathbb{H}$. Let $u_{ki}$ be the $i$th coordinate of $u_k$. We would like to find functions $\{\psi_k\} \subset \mathbb{H}$ such that $\psi_k(x_i) = \sqrt{n}\, u_{ki}$ for all $k, i \in [n]$. If $\dim(\mathbb{H}) = \infty$, there are in general many such functions for each $k$. We pick the one that minimizes the $\mathbb{H}$-norm:

$$\psi_k := \operatorname{argmin} \big\{ \|\psi\|_{\mathbb{H}} : \psi \in \mathbb{H}, \ S_{\boldsymbol{x}}(\psi) = u_k \big\}.$$

Under Assumption 1, the above problem has a unique solution (i.e., the interpolation problem $S_{\boldsymbol{x}}(\psi) = u$ has a unique minimum-norm solution for any $u \in \mathbb{R}^n$.). Since $\{u_k\}$ is an orthonormal basis of $\mathbb{R}^n$, by construction, $\langle \psi_k, \psi_\ell \rangle_n = 1\{k = \ell\}$. Consider the set of functions

$$\widetilde{\mathbb{H}} := \Big\{ \sum_{k=1}^{r} \alpha_k \psi_k \mid \alpha_1, \ldots, \alpha_r \in \mathbb{R} \Big\}$$

equipped with the inner product defined via $\langle \psi_k, \psi_\ell \rangle_{\widetilde{\mathbb{H}}} = \frac{1}{\mu_k} 1\{k = \ell\}$ and extended to the whole of $\widetilde{\mathbb{H}}$ by bilinearity. Equivalently, if $f = \sum_{k=1}^r \alpha_k \psi_k$ and $g = \sum_{k=1}^r \beta_k \psi_k$, define $\langle f, g \rangle_{\widetilde{\mathbb{H}}} = \sum_k \alpha_k \beta_k / \mu_k$. Then, $\widetilde{\mathbb{H}}$ is an $r$-dimensional RKHS, with reproducing kernel (Appendix A)

$$\widetilde{\mathbb{K}}(x, y) := \sum_{k=1}^r \mu_k \psi_k(x) \psi_k(y) \tag{5}$$

and by construction $\widetilde{\mathbb{H}} \subset \mathbb{H}$, although $\widetilde{\mathbb{H}}$ and $\mathbb{H}$ are equipped with different inner products.

## 3.1 TKRR is itself a valid KRR

We define the truncated kernel ridge regression (TKRR) estimator as the usual KRR estimator over $\widetilde{\mathbb{H}}$, that is,

$$\widetilde{f}_{r,\lambda} = \underset{f \in \widetilde{\mathbb{H}}}{\operatorname{argmin}} \ \frac{1}{n} \sum_{i=1}^n (y_i - f(x_i))^2 + \lambda \|f\|_{\widetilde{\mathbb{H}}}^2. \tag{6}$$

We have the following characterization of $\widetilde{f}_{r,\lambda}$:

**Proposition 1.** *Let* $\widetilde{K} = \sum_{k=1}^r \mu_k u_k u_k^T = (\frac{1}{n} \widetilde{\mathbb{K}}(x_i, x_j))$ *be the truncated kernel matrix and*

$$\widetilde{S}_{\boldsymbol{x}}^*(\omega) := \frac{1}{\sqrt{n}} \sum_{j=1}^n \omega_j \widetilde{\mathbb{K}}(\cdot, x_j), \tag{7}$$

*which is the adjoint of $S_{\boldsymbol{x}}$ as a map from $\widetilde{\mathbb{H}}$ to $\mathbb{R}^n$. Let $\widetilde{\Omega}$ be the solution set of* (3) *with $K$ replaced with $\widetilde{K}$. Then,* (6) *has a unique solution which is given by $\widetilde{f}_{r,\lambda} = \widetilde{S}_{\boldsymbol{x}}^*(\omega)$ for any $\omega \in \widetilde{\Omega}$.*

Although the above proposition is not the main contribution of this work, it provides the interesting observation that TKRR is an exact KRR estimator over a smaller RKHS. See also (21) in Appendix B for a closed-form expression for $\widetilde{f}_{r,\lambda}$.

## 4 Target alignment

We start with the definition of target alignment:

**Definition 1.** The alignment spectrum of a target function $f^*$ is $\xi^* = U^T S_{\boldsymbol{x}}(f^*) \in \mathbb{R}^n$ where $U$ is the orthogonal matrix of the eigenvectors of $K$, with columns corresponding to eigenvalues of $K$ in descending order. The elements of $\xi^*$ are referred to as target alignment (TA) scores.

The following theorem gives an exact expression for the expected empirical MSE of TKRR estimate, in terms of the TA scores $\xi^*$ and the eigenvalues $(\mu_i)$ of the kernel matrix:

**Theorem 1** (Exact MSE). *Let $\Gamma_\lambda$ be an $n \times n$ diagonal matrix with diagonal elements:*

$$(\Gamma_\lambda)_{ii} = \frac{\mu_i}{\mu_i + \lambda} 1\{1 \leq i \leq r\}. \tag{8}$$

*For any TKRR solution $\widetilde{f}_{r,\lambda}$, we have*

$$\mathbb{E}\|\widetilde{f}_{r,\lambda} - f^*\|_n^2 = \|(I_n - \Gamma_\lambda)\xi^*\|_2^2 + \frac{\sigma^2}{n} \operatorname{tr}(\Gamma_\lambda^2)$$

$$= \sum_{i=1}^r \frac{\lambda^2}{(\mu_i + \lambda)^2} (\xi_i^*)^2 + \sum_{i=r+1}^n (\xi_i^*)^2 + \frac{\sigma^2}{n} \sum_{i=1}^r \frac{\mu_i^2}{(\mu_i + \lambda)^2} \tag{9}$$

$$= \|f^*\|_n^2 + \sum_{i=1}^r \frac{1}{(\mu_i + \lambda)^2} \left[ -a_i(\lambda)(\xi_i^*)^2 + \frac{\sigma^2}{n} \mu_i^2 \right] \tag{10}$$

*where $a_i(\lambda) = (\mu_i + \lambda)^2 - \lambda^2$ and the expectation is w.r.t. the randomness in the noise vector $w$.*

Assume that the target function is normalized so that $\|f^*\|_n = 1$. Since $U$ is an orthogonal matrix, this is equivalent to $\|\xi^*\|_2 = 1$. Since the coefficient of $(\xi_i^*)^2$ in (10) is negative, the more the alignment spectrum peaks near the lower indices, the smaller the expected MSE. This observation is made more precise in Proposition 2 below.

## 4.1 Bandlimited alignment

To see the implications of Theorem 1, let us first consider a *bandlimited model* as follows: $f^*$ is randomly generated with alignment scores $\xi_i^*$ satisfying

$$\mathbb{E}(\xi_i^*)^2 = \frac{1}{b} 1\{\ell + 1 \leq i \leq \ell + b\} \tag{11}$$

for $\ell \geq 0$ and $b \geq 1$ (both integers). Then, we write $f^* \sim \mathcal{B}_{b,\ell}$ and note that $\mathbb{E}\|f^*\|_n^2 = 1$ for such $f^*$. We write $\overline{\text{MSE}} := \mathbb{E}\|\widetilde{f}_{r,\lambda} - f^*\|_n^2$ where the expectation is over the randomness in both $f^*$ and $w$—the noise vector in (1). One can think of this setting as a Bayesian model for $f^*$ and of $\overline{\text{MSE}}$ as the Bayes risk w.r.t. to the aforementioned prior on $f^*$. We leave the specification of the prior open subject to the condition (11) since this is the only assumption we need.

**Proposition 2.** *With $f^* \sim \mathcal{B}_{b,\ell}$, we have*

$$\overline{\text{MSE}} = 1 - \frac{1}{b} \sum_{i=\ell+1}^{(\ell+b)\wedge r} \frac{a_i(\lambda)}{(\mu_i + \lambda)^2} + \frac{\sigma^2}{n} \sum_{i=1}^{r} \frac{\mu_i^2}{(\mu_i + \lambda)^2}. \tag{12}$$

*Assume for simplicity that $\{\mu_i\}$ are distinct. Then, the following hold:*

(a) *For fixed $\ell$, $b$ and $\lambda$, let $j^* = \max\{i \in [\ell+1, \ell+b] \mid 1 + 2\lambda/\mu_i \leq \sigma^2 b/n\}$ if the set is nonempty, otherwise $j^* = \ell$. Then, the $\overline{\text{MSE}}$ as a function of $r$ is (i) increasing in $[1, j^*)$, (ii) decreasing in $[j^*, \ell+b]$, and (iii) increasing in $[\ell+b, n]$.*

(b) *For fixed $r$, $b$ and $\lambda$, the $\overline{\text{MSE}}$ as a function of $\ell$ is increasing in $\ell \in [0, r-b]$.*

(c) *For fixed $r$, $\ell$ and $\lambda$, the $\overline{\text{MSE}}$ as a function of $b$ is increasing in $b \in [1, r-\ell]$.*

Part (a) of Proposition 2 shows that $\overline{\text{MSE}}$ as a function of $r$ can be non-monotonic: go up, down and back up. This happens whenever $j^* < \ell + b$. Similar observations as Proposition 2(a) can be made about bandlimited signals consisting of multiple nonzero bands, e.g., $f^* = \frac{1}{\sqrt{K}}(f_1^* + \cdots + f_K^*)$ where $f_i^* \in \sim \mathcal{B}_{b_i, \ell_i}$ with non-overlapping bands. In this case, $\overline{\text{MSE}}$ can show multiple descents and ascents (cf. Figure 2a). Part (b) shows that alignment spectra that are concentrated near lower indices are better. Part (c) shows that concentrated alignment spectra are better than diffuse ones.

**Effect of noise level on optimal rank $r$**    Let us explore the implications of Proposition 2(a) for the optimal truncation level $r$ (optimal rank). Fix $\lambda > 0$. Then, for small enough noise levels (i.e., small $\sigma^2/n$), we have $1 + 2\lambda/\mu_{\ell+1} > \sigma^2 b/n$. Since $i \mapsto \mu_i$ is (strictly) decreasing, it follows that $j^* = \ell$, and $r \mapsto \overline{\text{MSE}}$ is decreasing over the entire interval $[\ell, \ell+b]$. That is, for $r < \ell + 1$, the $\overline{\text{MSE}}$ is increasing, but the moment we enter the signal band ($r = \ell + 1$), the $\overline{\text{MSE}}$ starts to decrease and keeps decreasing till we get to the end of the band ($r = \ell + b$), at which point it starts to increase again. For small enough noise levels, the descent within the band is big enough to bring the $\overline{\text{MSE}}$ below its level at $r = 1$; hence the optimal truncation level, in this case, is at the end of the band: $r = \ell + b$. If we increase the noise level, at some point, $j^*$ starts to move to the middle of the band $[\ell, \ell+b]$. In this case, depending on where in the band $j^*$ lies, the optimal truncation level could still be $r = \ell + b$. But as we increase the noise level, there comes a point where the noise level is so high that $j^*$ is very close to the end of the band, in which case, the dip in the $\overline{\text{MSE}}$ is not big enough to bring it below its $r = 1$ level. Then, the best truncation is just at $r = 1$ level. This includes the case where $j^* = \ell + b$ in which case the $\overline{\text{MSE}}$ is monotonically increasing (very high noise levels). Similar ideas play out in the multi-band setting as shown in Figure 2a in Section 5.

## 4.2 Polynomial alignment

We now consider the case where the TA scores decay polynomially with a rate potentially faster or slower than what is required by merely belonging to the RKHS. In particular, assume that

$$\mu_i \asymp i^{-\alpha}, \quad (\xi_i^*)^2 \asymp i^{-2\gamma\alpha - 1} \tag{13}$$

for some $\gamma > 0$ and $\alpha \geq 1$. Let us justify the choice of decay for $(\xi_i^*)^2$. Asymptotically, both $(\xi_i^*)^2$ and $\mu_i$ should decay similar to the population level TA scores and eigenvalues. Then, $f^* \in \mathbb{H}$, if and

only if $\sum_i (\xi_i^*)^2/\mu_i \lesssim 1$ (see Appendix D). Assuming a polynomial decay for $(\xi_i^*)^2$, it follows that that $(\xi_i^*)^2$ should decay faster than $i^{-\alpha-1}$ for the sum to converge, that is $\lesssim i^{-\alpha-1-\delta}$ for some $\delta > 0$. Without loss of generality, we are assuming $\delta = (2\gamma - 1)\alpha$. This parameterization is chosen to be consistent with the existing literature. Thus, for a function $f^* \in \mathbb{H}$, $\xi^*$ satisfies (13) with $\gamma > 1/2$. The case $\gamma \leq 1/2$ then corresponds to a target that does not belong to the RKHS.

The following result characterizes the performance of full KRR and TKRR under model (13):

**Theorem 2.** *Let $\eta = \min(r, \lambda^{-1/\alpha})$. Under the polynomial decay assumption* (13)*,*

$$\mathbb{E}\|\widetilde{f}_{r,\lambda} - f^*\|_n^2 \; \asymp \; \lambda^2 \max(1, \eta^{-2(\gamma-1)\alpha}) + r^{-2\gamma\alpha}1\{r < n\} + \frac{\sigma^2}{n}\eta. \tag{14}$$

*(a) Taking $\lambda \asymp (\sigma^2/n)^{\gamma\alpha/(2\gamma\alpha+1)}$ and $r \asymp (n/\sigma^2)^{1/(2\gamma\alpha+1)}$, TKRR achieves the following rate*

$$\mathbb{E}\|\widetilde{f}_{r,\lambda} - f^*\|_n^2 \; \asymp \; \left(\frac{\sigma^2}{n}\right)^{2\gamma\alpha/(2\gamma\alpha+1)} \quad \text{for } \gamma > 1. \tag{15}$$

*(b) Assume $n^{-2\alpha} \lesssim \sigma^2 \lesssim n$, and let $\delta := \min(1, \gamma)$. Then, the best rate achievable by the full KRR is obtained for regularization choice $\lambda \asymp (\sigma^2/n)^{\alpha/(2\delta\alpha+1)}$ and is given by*

$$\mathbb{E}\|\widetilde{f}_{n,\lambda} - f^*\|_n^2 \; \asymp \; \left(\frac{\sigma^2}{n}\right)^{2\delta\alpha/(2\delta\alpha+1)} \quad \text{for } \gamma > 0. \tag{16}$$

Note that the exponent $\delta$ saturates to $\delta = 1$ for all $\gamma > 1$ giving the best rate $(\sigma^2/n)^{2\alpha/(2\alpha+1)}$ for the full KRR in those cases. In contrast, the rate in (15) achievable by TKRR improves for $\gamma > 1$ without bound, and in the limit of $\gamma \to \infty$, approaches $\sigma^2/n$, the best parametric rate.

It is also well-known that the minimax rate over the unit ball of the RKHS with eigendecay $\mu_i \asymp i^{-\alpha}$ is $(\sigma^2/n)^{\alpha/(\alpha+1)}$. This can also be seen by letting $\gamma \to 1/2$ in either (15) or (16), removing any target alignment assumption beyond what is provided by the RKHS itself.

To summarize, let us define the rate exponent function

$$s(\gamma) := 2\gamma\alpha/(2\gamma\alpha + 1). \tag{17}$$

There are four regimes of target alignment, implied by Theorem 2:

1. Under-aligned regime, $\gamma \in (0, \frac{1}{2})$: The target is not in the RKHS. The best achievable rate is $(\sigma^2/n)^{s(\gamma)}$ which is slower than the minimax rate over the ball of the RKHS, $(\sigma^2/n)^{s(\frac{1}{2})}$.

2. Just-aligned regime, $\gamma = \frac{1}{2}$ (to be precise $\gamma \downarrow \frac{1}{2}$): The target is only assumed to be in the RKHS. The best rate achievable is the minimax rate over the ball of the RKHS, $(\sigma^2/n)^{s(\frac{1}{2})}$.

3. Weakly-aligned regime, $\gamma \in (\frac{1}{2}, 1]$: The target is in the RKHS and more aligned with the kernel than what is implied by being in the RKHS. The best achievable rate is $(\sigma^2/n)^{s(\gamma)}$ which is faster than the minimax rate over the ball of the RKHS, $(\sigma^2/n)^{s(\frac{1}{2})}$. The rate is achievable by the full KRR and hence TKRR.

4. Over-aligned regime, $\gamma > 1$: The target is in the RKHS and is strongly aligned with the kernel. The best achievable rate is $(\sigma^2/n)^{s(\gamma)}$ which is achieved by TKRR. The full KRR can only achieve the rate $(\sigma^2/n)^{s(1)}$ in this case, which is the best achievable in the weakly-aligned regime.

Note that the best achievable rate in the first three regimes is attainable by full KRR (hence TKRR) while for the fourth regime, only TKRR can achieve the given rate. It was shown in [9] (cf. Appendix D.2 for details) that the rate given in the weakly-aligned regime is minimax optimal over the class of targets (13) with $\gamma \in (1/2, 1]$. We conjecture that the minimax optimality of this rate extends to the over-aligned regime, that is, TKRR is minimax optimal for all $\gamma > 1/2$.

The capped rate (16) for the full KRR is the same as the one obtained in [12, Eqn. (12)]. The rate (15) for the TKRR in the over-aligned regime is new to the best of our knowledge. The work of [2] shows some improvement in TKRR over full KRR, but since the minimax rate was considered only over the unit ball, no difference in rates was shown between TKRR and full KRR. In contrast, by

considering the smaller alignment class (13), we can show the rate differences. We also note that the cross-over effects noted in [12] for different regularization regimes follow easily from the simplified formula (14), as can be seen by inspecting the various cases in the proof of part (b) of Theorem 2. To simplify the statement of Theorem 2, we have focused only on the optimal regularization regime.

There are interesting parallels but also notable differences between our work and that of [25]. They consider the usual ridge regression while we work with the kernel ridge, although the results can be translated back and forth after some transformation. They use a spectral truncation parameter $k$, as a proof-device to obtain the tightest upper bound, whereas the truncation level $r$ in our case is a regularization parameter in TKRR that can be tuned in practice. The bound in [25] essentially corresponds to that of the Full KRR in our paper. In fact, we also use a similar proof-device $k$ in the proof of Theorem 2, separate from the $r$-truncation level of TKRR. As an important corollary, our results about TKRR cannot be deduced from those in [25]. Another point of difference is the focus in [25] which is on how the eigendecay of the kernel (or equivalently the covariance matrix) affects the MSE bound, while our focus is on how the interaction of the target alignment decay and the kernel eigendecay together affect the bound. There are indications of the effect of the target alignment in [25], in the form of the tail energy of their $\theta^*$ parameter, but the implications of this decay seems to have not been explored in detail compared to our work.

**Generalization error** We have considered a fixed design so far, that is, fixed $(x_i)$. To see the connections with the generalization error, we need to consider a random design. Let $(x, y)$ be a random test point, and let $(x_1, y_1), \ldots, (x_n, y_n)$ be i.i.d. training data, all from the same joint distribution $\mathbb{P}$ on $(x, y)$. Let $\mathbb{P}_X$ be the marginal distribution of $x$ under $\mathbb{P}$, and for a function $f$, write $\|f\|_{L^2}^2 = \int f^2(x) d\mathbb{P}_X(x)$ for the corresponding (population) squared $L^2$ norm. Recalling that $y = f^*(x) + \varepsilon$ and $\mathbb{E}[\varepsilon] = 0$, the generalization error of a deterministic $f$ is

$$\mathbb{E}(y - f(x))^2 = \mathbb{E}(f^*(x) - f(x))^2 + \sigma^2 = \|f - f^*\|_{L^2}^2 + \sigma^2,$$

where the first expectation is taken w.r.t. the randomness in both $x$ and $y$ (that is, $x$ and $\varepsilon$). The noise variance $\sigma^2$ is the unimprovable part of the generalization error, i.e., the minimum Bayes risk. So, the excess generalization error is $\|f - f^*\|_{L^2}^2$. For large $n$, since the $L^2$ norm is an integral, it can be well-approximated by an average based on the training $\boldsymbol{x} = (x_i)_{i=1}^n$, that is, $\frac{1}{n}\sum_{i=1}^n (f(x_i) - f^*(x_i))^2$ which is the empirical norm that we have considered in this paper. In fact, letting $B_n$ be the expression in (10) and taking the expectation of both sides of (10) w.r.t. $\boldsymbol{x}$, by Fubini theorem, we have $\mathbb{E}\|\widetilde{f}_{r,\lambda} - f^*\|_{L^2}^2 = \mathbb{E}_{\boldsymbol{x}}(B_n)$ where $\mathbb{E}_{\boldsymbol{x}}$ denotes the expectation w.r.t. training $\boldsymbol{x}$. This controls the expected excess generalization error of $\widetilde{f}_{r,\lambda}$.

More precise high-probability bounds on $\|\widetilde{f}_{r,\lambda} - f^*\|_{L^2}^2$ could be possible, by techniques similar to those in [28, Theorem 14.12], establishing a high probability uniform control of the form $\|f\|_{L^2}^2 \leq 2\|f\|_n^2$ over the unit ball of an RKHS generated by a Mercer kernel. There are, however, technical difficulties in controlling the so-called critical radius in these arguments at the level needed for the fast rates we present here. Combined with a sub-Gaussian assumption on the noise vector $\varepsilon$, these arguments could lead to high probability bounds on the population excess generation error similar to the results we present here. Whether such arguments are feasible is left for future work.

## 5 Simulations

We present various simulation results to demonstrate the multiple-descent and phase transition behavior of the regularization curves, and corroborate the theoretical results. First, consider the bandlimited alignment (11) where the nonzero entries of $\xi^*$ are i.i.d. draws from $N(0, 1)$ and $\xi^*$ is normalized to have unit norm. We use the Gaussian kernel $e^{-\|x-x'\|/2h^2}$ in $d = 4$ dimensions with bandwidth $h = \sqrt{d/2}$, applied to 200 samples generated from a uniform distribution on $[0, 1]^d$, and let $r = l + b$.

The plots in Figure 1a show the multiple-descent and phase transition behavior of the $\lambda$-regularization curves (that is, expected MSE versus $-\log(\lambda)$) for different values of the noise level $\sigma$. The corresponding contour plot is shown in Figure 1b. The plots show a transition from monotonically decreasing at $\sigma = 0$ to monotonically increasing for large $\sigma$, with non-monotonic behavior for moderate values of $\sigma$.

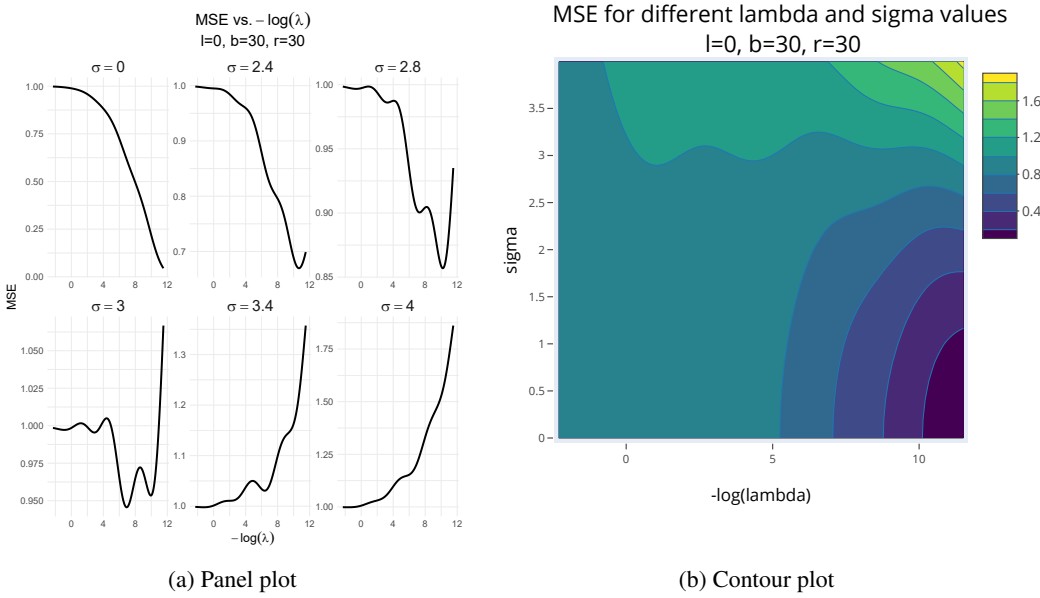

(a) Panel plot

(b) Contour plot

Figure 1: Multiple-descent and phase transition of for $\lambda$-regularization curve: (a) Expected MSE as a function of $-\log(\lambda)$ for different values of $\sigma$, and (b) overall contour plot of expected MSE for $\sigma$ vs. $-\log(\lambda)$.

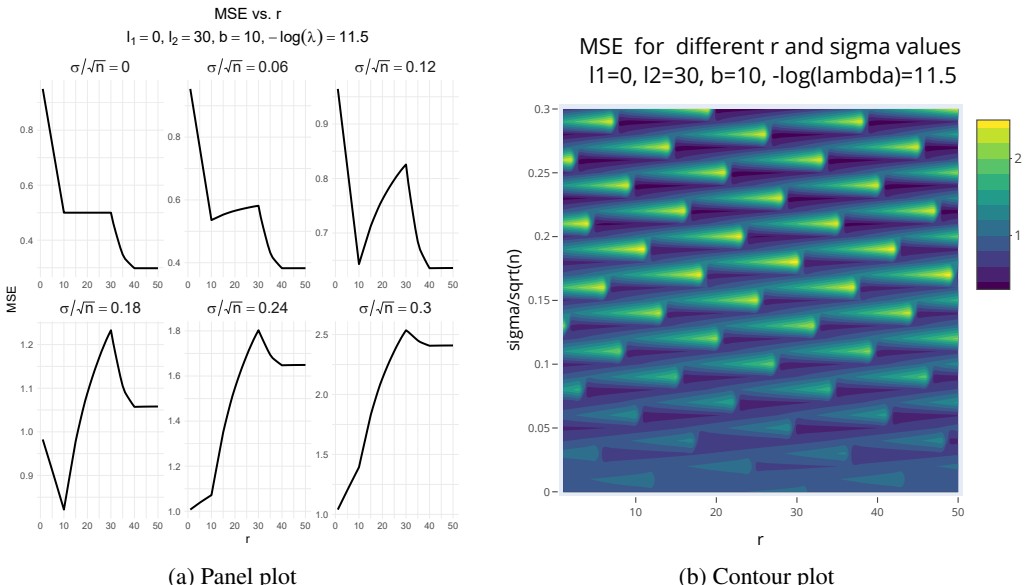

(a) Panel plot

(b) Contour plot

Figure 2: Double-descent and phase transition for $r$-regularization curves: (a) Expected MSE as a function of the truncation parameter $r$. Plots are indexed by different values of $\sigma/\sqrt{n}$, and the $x$-axis starts at $r = 1$. (b) Overall contour plot of expected MSE for $\sigma/\sqrt{n}$ vs. $r$.

Next, we consider a bandlimited alignment with two non-overlapping bands of length $b$, starting at indices $\ell_1 + 1$ and $\ell_2 + 1$. The nonzero entries of $\xi^*$ are generated as before, followed by unit norm normalization. We fix $\lambda$ and consider the $r$-regularization curve, that is, the plot of MSE versus truncation parameter $r$. The results are shown in Figure 2a for different values of $\sigma/\sqrt{n}$. Figure 2b shows the corresponding contour plot. The plots show the double-descent and phase transition behavior of the $r$-regularization curves. The double-decent behavior is consistent with Proposition 2(a) that predicts the MSE goes up as $r$ varies between bands. Simulations corroborating Theorem 2 are provided in Appendix C. The code for reproducing the simulations is available at [3].

# 6  Conclusion

We presented an analysis of the TKRR in the light of recent advances that were made for the KRR. We showed the TKRR equivalence to a KRR on a (smaller) RKHS embedded in the original RKHS. We believe that this insight provides a different perspective on the TKRR and unifies it with the KRR development. We derived an exact MSE decomposition for the TKRR via TA scores and eigenvalues of the kernel matrix. When TA scores exhibit polynomial decay we identified an *over-aligned* regime, where the TKRR can outperform ordinary KRR. When the TA scores are bandlimited, we demonstrated the non-monotonicity of the regularization curve with respect to the level of truncation and a correspondence of narrower TA spectra that are positioned towards lower indices with lower generalization error.

Although we defined the TA spectrum based on the true $f^*$, there is a very good plugin estimator for it in practice: Replacing $S_{\boldsymbol{x}}(f^*)$ with the noisy observations $y/\sqrt{n}$ in the definition of the spectrum gives a very good estimate, that is, $U^T y/\sqrt{n}$ is a very good estimate of $U^T S_{\boldsymbol{x}}(f^*)$. This is because each $u_k^T y/\sqrt{n}$ will effectively average over the noise in $y$.

Our results can be easily extended to the non-additive and heteroscedastic noise setting, where $(x, y)$ are drawn from a general distribution, $f^*(x) := \mathbb{E}[y \mid x]$ and $\varepsilon = y - f^*(x)$. The extension is possible if we assume $\mathrm{var}(\varepsilon \mid x) \leq \sigma^2$, in which case, Theorem 1, for example, holds as an upper bound on the MSE. However, extensions to more general loss functions, such as logistic loss for classification, requires additional work. Whether the improved rate (15) for TKRR is minimax optimal for $\gamma > 1$, appears to be open, although we conjecture it to be the case. Due to the theoretical nature of this contribution, no negative societal impact of our work is anticipated.

# 7  Proofs

Here, we only prove Theorem 2. The remaining proofs are given in Appendix B.

*Proof of Theorem 2.* To simplify the argument, we assume that $\lambda^{-1/\alpha}$ is an integer, without loss of generality. Recall that for a decreasing function $f$, we have $\sum_{i=L}^{U} f(i) \leq \int_{L-1}^{U} f(x)dx$. Let us first consider the sum involved in the variance term. We will use the following inequalities, $\frac{1}{2}\min(a, b) \leq \frac{ab}{a+b} \leq \min(a, b)$ which holds for $a, b \geq 0$. We have

$$I_1 := \sum_{i=1}^{r} \frac{\mu_i^2}{(\mu_i + \lambda)^2} = \frac{1}{\lambda^2} \sum_{i=1}^{r} \frac{\mu_i^2 \lambda^2}{(\mu_i + \lambda)^2} \asymp \frac{1}{\lambda^2} \sum_{i=1}^{r} \min(\lambda^2, \mu_i^2).$$

Since $\mu_i = i^{-\alpha}$, the minimum above is $\lambda^2$ for $i \in [1, \lambda^{-1/\alpha}]$, and $\mu_i^2$ for $i > \lambda^{-1/\alpha}$. If $r \leq \lambda^{-1/\alpha}$, only the first case happens and the overall bound is $\frac{1}{\lambda^2}r\lambda^2 = r$. If $r > \lambda^{-1/\alpha} =: k$, we have

$$I_1 \asymp \frac{1}{\lambda^2}\left(k\lambda^2 + \sum_{i=k+1}^{r} i^{-2\alpha}\right) = k + \frac{1}{\lambda^2}\int_k^r x^{-2\alpha}dx \lesssim k + \lambda^{-2}k^{-2\alpha+1} \lesssim \lambda^{-1/\alpha}$$

To summarize, $I_1 \lesssim \min(r, \lambda^{-1/\alpha})$ and we note that the variance term in (9) is $(\sigma^2/n)I_1$.

For the middle term in (9), we have, assuming $r < n$,

$$\sum_{i=r+1}^{n} (\xi_i^*)^2 = \sum_{i=r+1}^{n} i^{-2\gamma\alpha-1} \leq \int_r^\infty x^{-2\gamma\alpha-1}dx \lesssim r^{-2\gamma\alpha}.$$

Finally, for the first term in (9), we can write

$$I_2 := \sum_{i=1}^{r} \frac{\lambda^2}{(\mu_i + \lambda)^2}(\xi_i^*)^2 = \sum_{i=1}^{r} \frac{\lambda^2 \mu_i^2}{(\mu_i + \lambda)^2}\left(\frac{\xi_i^*}{\mu_i}\right)^2 \asymp \sum_{i=1}^{r} \min(\lambda^2, \mu_i^2)\left(\frac{\xi_i^*}{\mu_i}\right)^2.$$

If $r \leq \lambda^{-1/\alpha}$, we have $I_2 \leq \lambda^2 \sum_{i=1}^{r}\left(\frac{\xi_i^*}{\mu_i}\right)^2 \asymp \lambda^2 \sum_{i=1}^{r} i^{-2(\gamma-1)\alpha-1}$. Letting $\beta = 2(\gamma-1)\alpha$,

$$\frac{1}{\lambda^2}I_2 \leq 1 + \sum_{i=2}^{r} i^{-\beta-1} \leq 1 + \int_1^r x^{-\beta-1}dx = 1 + \frac{1}{\beta}(1 - r^{-\beta}).$$

If $\gamma \geq 1$, the sum above is $\lesssim 1$. If $\gamma < 1$, then $\beta < 0$ and the sum is $\lesssim r^{|\beta|}$. The two cases can be compactly written as $\lesssim \max(1, r^{-\beta})$. If $r > \lambda^{-1/\alpha} := k$, we have

$$I_2 \;\leq\; \lambda^2 \sum_{i=1}^{k} \Big(\frac{\xi_i^*}{\mu_i}\Big)^2 + \sum_{i=k+1}^{r} (\xi_i^*)^2 \;\lesssim\; \lambda^2 \max(1, k^{-\beta}) + k^{-2\gamma\alpha}$$

using the bounding techniques we have seen before. Plugging in $k = \lambda^{-1/\alpha}$, we obtain

$$I_2 \;\lesssim\; \lambda^2 \max(1, \lambda^{2(\gamma-1)}) + \lambda^{2\gamma} \;\lesssim\; \max(\lambda^2, \lambda^{2\gamma})$$

To summarize, for $r \leq \lambda^{-1/\alpha}$, we have $I_2 \lesssim \lambda^2 \max(1, r^{-\beta})$ while for $r > \lambda^{-1/\alpha}$, we have $I_2 \lesssim \lambda^2 \max(1, \lambda^{2(\gamma-1)})$. Note that these two expressions can be combined into one as follows:

$$I_2 \;\lesssim\; \lambda^2 \max(1, \eta^{-\beta}),$$

where $\eta = \min(r, \lambda^{-1/\alpha})$. Putting the pieces together gives the desired bound. The above upper bounds are sharp up to constants, since in each step, there is a corresponding matching lower bound.

**Proof of part (a).** With the given choice for $\lambda$ and $r$, we have $\lambda^{-1/\alpha} \asymp (n/\sigma^2)^{\gamma/(2\gamma\alpha+1)} \geq (n/\sigma^2)^{1/(2\gamma\alpha+1)} \asymp r$ for $\gamma \geq 1$. Thus $\eta = \min(r, \lambda^{-1/\alpha}) \asymp r$, hence $\eta^{-2(\gamma-1)\alpha} \lesssim 1$ for $\gamma \geq 1$ since $r \geq 1$. It follows that the

$$\mathbb{E}\|\widetilde{f}_{r,\lambda} - f^*\|_n^2 \;\asymp\; \lambda^2 + r^{-2\gamma\alpha} + \frac{\sigma^2}{n}r.$$

Plugging in the choices for $\lambda$ and $r$, each of the three terms is $\asymp (\sigma^2/n)^{2\gamma\alpha/(2\gamma\alpha+1)}$ as desired.

**Proof of part (b).** Here, we have $r = n$, so the middle term in (14) vanishes. Assume first that $\gamma \geq 1$. We consider different regularization regimes.

Case 1: $\lambda^{-1/\alpha} \in [1, n]$ so that $\lambda \in [n^{-\alpha}, 1]$ and $\eta = \min(n, \lambda^{-1/\alpha}) = \lambda^{-1/\alpha}$. We have $\eta^{-2(\gamma-1)\alpha} = \lambda^{2(\gamma-1)} \leq 1$. It follows that

$$\mathbb{E}\|\widetilde{f}_{r,\lambda} - f^*\|_n^2 \;\asymp\; \lambda^2 + \frac{\sigma^2}{n}\lambda^{-1/\alpha}.$$

The optimal choice of $\lambda$ is obtained by setting $\lambda^2 \asymp (\sigma^2/n)\lambda^{-1/\alpha}$, that is, $\lambda = (\sigma^2/n)^{\alpha/(2\alpha+1)}$ and MSE rate $\asymp (\sigma^2/n)^{2\alpha/(2\alpha+1)}$. Note that, assuming $n^{-2\alpha} \lesssim \sigma^2 \lesssim n$, this choice of $\lambda$ is within the assumed range $[n^{-\alpha}, 1]$ up to constants.

Case 2: $\lambda^{-1/\alpha} < 1$. Then, $\eta = 1$ and the MSE $\asymp \lambda^2 + \sigma^2/n$. Since $\sigma^2 \lesssim n$ by assumption and $\lambda > 1$, we have MSE $\asymp \lambda^2 > 1$ which is always worse than the rate in Case 1.

Case 3: $\lambda^{-1/\alpha} > n$. Then, $\eta = n$ and MSE $\asymp \lambda^2 + \sigma^2 \asymp \sigma^2$ since $\sigma^2 \gtrsim n^{-2\alpha}$ by assumption and $\lambda^2 < n^{-2\alpha}$. This rate is clearly worst than Case 1. Putting the pieces together the optimal rate is achieved by Case 1.

Next, consider the case $\gamma < 1$. We again consider the three regularization regimes:

Case 1: $\lambda^{-1/\alpha} \in [1, n]$ so that $\lambda \in [n^{-\alpha}, 1]$ and $\eta = \min(n, \lambda^{-1/\alpha}) = \lambda^{-1/\alpha}$. We have $\eta^{-2(\gamma-1)\alpha} = \lambda^{2(\gamma-1)} > 1$. It follows from (14) that

$$\mathbb{E}\|\widetilde{f}_{r,\lambda} - f^*\|_n^2 \;\asymp\; \lambda^{2\gamma} + \frac{\sigma^2}{n}\lambda^{-1/\alpha}.$$

The optimal choice of $\lambda$ is obtained by setting $\lambda^{2\gamma} \asymp (\sigma^2/n)\lambda^{-1/\alpha}$, that is, $\lambda = (\sigma^2/n)^{\alpha/(2\gamma\alpha+1)}$ and MSE rate $\asymp (\sigma^2/n)^{2\gamma\alpha/(2\gamma\alpha+1)}$. Note that, as long as $n^{-2\gamma\alpha} \lesssim \sigma^2 \lesssim n$ (which holds by the assumption on $\sigma^2$ since $\gamma < 1$), this choice of $\lambda$ is within the assumed range $[n^{-\alpha}, 1]$ up to constants.

Case 2: $\lambda^{-1/\alpha} < 1$. This is similar to Case 2 when $\gamma \geq 1$.

Case 3: $\lambda^{-1/\alpha} > n$. Then, $\eta = n$ and $\eta^{-2(\gamma-1)\alpha} > 1$, hence MSE $\asymp \lambda^2 n^{-2(\gamma-1)\alpha} + \sigma^2 \gtrsim \sigma^2$ which is clearly worst than the rate in Case 1. Putting the pieces together the optimal rate is achieved by Case 1. $\qquad\square$

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
