# Appendix

This appendix contains the remaining proofs, additional simulations, some background on the RKHS and the connections between different settings in the literature on kernel ridge regression.

## A  Verifying reproducing property

To see the reproducing property of the kernel $\widetilde{\mathbb{K}}$, defined in (5), for the space $\widetilde{\mathbb{H}}$, note that for any $f = \sum_\ell \alpha_\ell \psi_\ell \in \widetilde{\mathbb{H}}$, we have

$$\langle f, \widetilde{\mathbb{K}}(\cdot, y)\rangle_{\widetilde{\mathbb{H}}} = \sum_{\ell=1}^r \sum_{k=1}^r \alpha_\ell \mu_k \psi_k(y) \langle \psi_\ell, \psi_k \rangle_{\widetilde{\mathbb{H}}} = \sum_{k=1}^r \alpha_k \psi_k(y) = f(y).$$

## B  Remaining proofs

*Proof of Proposition 1.* We write $y = (y_1, \ldots, y_n)$ and $\varepsilon = (\varepsilon_1, \ldots, \varepsilon_n)$. Then, the model can be compactly written as $y = \sqrt{n} S_{\boldsymbol{x}}(f^*) + \varepsilon$. Let $\widetilde{y} = y/\sqrt{n}$ and $\widetilde{\varepsilon} = w/\sqrt{n}$ so that

$$\widetilde{y} = S_{\boldsymbol{x}}(f^*) + \widetilde{\varepsilon}.$$

By the representer theorem, a general TKRR solution is $\widetilde{f}_{r,\lambda} = \widetilde{S}_{\boldsymbol{x}}^*(\widetilde{\omega})$ where $\widetilde{\omega}$ is a solution of

$$\min_{\omega \in \mathbb{R}^n} \frac{1}{n}\|y - \sqrt{n}\widetilde{K}\omega\|^2 + \lambda \omega^T \widetilde{K}\omega. \tag{18}$$

The first-order optimality condition gives $\widetilde{K}[(\widetilde{K}\widetilde{\omega} - \widetilde{y}) + \lambda\widetilde{\omega}] = 0$. Let us write $K = U\Lambda U^T$ for the eigen-decomposition of the full kernel matrix, where $\Lambda = \mathrm{diag}(\mu_1, \ldots, \mu_n)$ and $U$ has columns $u_1, \ldots, u_n$. Let $U_1 = [u_1 \mid u_2 \mid \cdots \mid u_r] \in \mathbb{R}^{n \times r}$, $U_2 = [u_{r+1} \mid \cdots \mid u_n] \in \mathbb{R}^{n \times (n-r)}$ and $\Lambda_1 = \mathrm{diag}(\mu_1, \ldots, \mu_r) \in \mathbb{R}^{r \times r}$. Then, $\widetilde{K} = U_1 \Lambda_1 U_1^T$ and $\widetilde{\omega} = U_1\alpha + U_2\beta$ for some vectors $\alpha$ and $\beta$. Substituting into the first-order condition and noting $U_1^T U_1 = I_r$ and $U_1^T U_2 = 0$, we have

$$U_1\Lambda_1[(\Lambda_1\alpha - U_1^T\widetilde{y}) + \lambda\alpha] = 0.$$

Let $\xi_{(1)} = U_1^T\widetilde{y}$. Multiplying both sides of the above by $\Lambda_1^{-1}U_1^T$, we obtain $(\Lambda_1\alpha - \xi_{(1)}) + \lambda\alpha = 0$. Letting $A_\lambda = \Lambda_1 + \lambda I_r$, we have $\alpha = A_\lambda^{-1}\xi_{(1)}$. Thus all the solutions $\widetilde{\omega}$ of (18) are of the form

$$\widetilde{\omega} = U_1 A_\lambda^{-1}\xi_{(1)} + U_2\beta \tag{19}$$

for an arbitrary $\beta \in \mathbb{R}^{n \times (n-r)}$.

Next, combining (5) and (7), we have for any $\omega \in \mathbb{R}^n$,

$$\widetilde{S}_{\boldsymbol{x}}(\omega) = \frac{1}{\sqrt{n}} \sum_{j=1}^n \omega_j \sum_{k=1}^r \mu_k \psi_k \psi_k(x_j) = \sum_{j=1}^n \omega_j \sum_{k=1}^r \mu_k \psi_k u_{kj} = \sum_{k=1}^r (u_k^T\omega)\mu_k\psi_k. \tag{20}$$

Since $u_k^T U_2\beta = 0$ for all $k = 1, \ldots, r$, it follows that $\widetilde{f}_{r,\lambda} = \widetilde{S}_{\boldsymbol{x}}^*(\widetilde{\omega})$ is the same regardless of the value of $\beta$, proving the uniqueness. In fact, noting that $u_k^T\widetilde{\omega} = (\mu_k + \lambda)^{-1}u_k^T\widetilde{y}$ for all $k \in [r]$,

$$\widetilde{f}_{r,\lambda} = \sum_{k=1}^r \frac{\mu_k}{\mu_k + \lambda}(u_k^T\widetilde{y})\psi_k. \tag{21}$$

For future reference, $\widetilde{f}_{r,\lambda} = \widetilde{S}_{\boldsymbol{x}}^*(\widetilde{\omega})$ implies $\widetilde{f}_{r,\lambda}(x_i) = \frac{1}{\sqrt{n}}\sum_{j=1}^n \widetilde{\omega}_j\widetilde{\mathbb{K}}(x_i, x_j) = \sqrt{n}[\widetilde{K}\widetilde{\omega}]_i$, hence

$$S_{\boldsymbol{x}}(\widetilde{f}_{r,\lambda}) = \widetilde{K}\widetilde{\omega} = (U_1\Lambda_1 U_1^T)(U_1 A_\lambda^{-1}\xi_{(1)} + U_2\beta) = U_1\Lambda_1 A_\lambda^{-1}\xi_{(1)}. \tag{22}$$

The proof is complete. $\qquad\square$

*Proof of Theorem 1.* Using the previous notation and recalling that $\xi^* = U^T S_{\boldsymbol{x}}(f^*)$, we have $\xi := U^T \widetilde{y} = \xi^* + \boldsymbol{z}$ and $\boldsymbol{z} := U^T \widetilde{\varepsilon}$. Writing $\xi = U^T \widetilde{y} = (\xi_{(1)}, U_2^T \widetilde{y})$, we can rewrite (22) as $S_{\boldsymbol{x}}(\widetilde{f}_{r,\lambda}) = U\Gamma_\lambda \xi$. It follows that

$$\|\widetilde{f}_{r,\lambda} - f^*\|_n^2 = \|S_{\boldsymbol{x}}(\widetilde{f}_{r,\lambda}) - S_{\boldsymbol{x}}(f^*)\|_2^2$$
$$= \|U\Gamma_\lambda \xi - U\xi^*\|_2^2$$
$$= \|\Gamma_\lambda \xi - \xi^*\|_2^2 = \|(\Gamma_\lambda - I_n)\xi^* + \Gamma_\lambda \boldsymbol{z}\|_2^2.$$

Expanding and using $\mathbb{E}[\boldsymbol{z}] = 0$, we get

$$\mathbb{E}\|\widetilde{f}_{r,\lambda} - f^*\|_n^2 = \|(I_n - \Gamma_\lambda)\xi^*\|_2^2 + \mathrm{tr}\big(\Gamma_\lambda^2 \mathbb{E}[\boldsymbol{z}\boldsymbol{z}^T]\big).$$

Noting that $\mathbb{E}[\boldsymbol{z}\boldsymbol{z}^T] = \mathrm{cov}(\boldsymbol{z}) = U^T \mathrm{cov}(\widetilde{\varepsilon})U = \frac{\sigma^2}{n} U^T U = \frac{\sigma^2}{n} I_n$ gives

$$\mathbb{E}\|\widetilde{f}_{r,\lambda} - f^*\|_n^2 = \|(I_n - \Gamma_\lambda)\xi^*\|_2^2 + \frac{\sigma^2}{n} \mathrm{tr}(\Gamma_\lambda^2)$$

which is the desired result. The expression (10) is obtained by writing $\sum_{i=r+1}^n (\xi_i^*)^2 = \|\xi^*\|_2^2 - \sum_{i=1}^r (\xi_i^*)^2$ and noting that $\|f\|_n = \|\xi^*\|_2$. $\qquad\square$

*Proof of Proposition 2.* The expression for $\overline{\mathrm{MSE}}$ follows by taking the expectation of both sides of (10) and noting that $\mathbb{E}(\xi_i^*)^2 = 1/b$ when nonzero and $\mathbb{E}\|f^*\|_n^2 = 1$.

For part (a), first we note that $\overline{\mathrm{MSE}}$ is increasing in intervals $[1, \ell]$ and $[\ell + b, n]$. This is immediate from the expression, since in both cases, the middle term in (12) remains constant as a function of $r$, while the the estimation error (the third term) contributes positive terms to the $\overline{\mathrm{MSE}}$ when increasing $r$. For the middle interval $[\ell, \ell + b]$, we consider the two intervals $[\ell, j^*] = [\ell, j^* - 1]$ and $[j^*, \ell + b]$ separately.

Assume first that $j^* \in [\ell + 1, \ell + b)$. Since $i \mapsto \mu_i$ is decreasing, we have

$$1 + \frac{2\lambda}{\mu_i} < \frac{\sigma^2}{n} b \quad \text{for } i \in [\ell + 1, j^* - 1],$$
$$1 + \frac{2\lambda}{\mu_i} \le \frac{\sigma^2}{n} b \quad \text{for } i = j^*,$$
$$1 + \frac{2\lambda}{\mu_i} > \frac{\sigma^2}{n} b \quad \text{for } i \in [j^* + 1, \ell + b]. \tag{23}$$

The first two lines above follow since $i \mapsto \mu_i$ is an strictly decreasing sequence by the distinctness of $\{\mu_i\}$, hence the inequality can potentially turn into an equality only at the endpoint $i = j^*$. The third line, (23), follows by the maximally of $j^*$. Inequality (23) is equivalent to $\frac{1}{b} a_i(\lambda) > \frac{\sigma^2}{n} \mu_i^2$ showing that the combined contribution to the $\overline{\mathrm{MSE}}$ by the $i$th terms of the two sums in (12) is negative for $i \in [j^* + 1, \ell + b]$, hence the $\overline{\mathrm{MSE}}$ is decreasing on $[j^*, \ell + b]$. Similarly, the first inequality shows that the combined contribution by the $i$th terms of the two sums in (12) is positive for $i \in [\ell + 1, j^* - 1]$, hence the $\overline{\mathrm{MSE}}$ is increasing in $[\ell, j^* - 1]$. This completes the proof of part (a) when $j^* \in [\ell + 1, \ell + b)$. Note that in this case, the $\overline{\mathrm{MSE}}$ is possibly flat only on $[j^* - 1, j^*]$. When $j^* = \ell + b$, the assertion about the interval $[j^*, \ell + b]$ is vacuous. When, $j^* = \ell$, by definition, inequality (23) holds for all $i \in [\ell + 1, \ell + b]$, hence the $\overline{\mathrm{MSE}}$ is decreasing in $[\ell, \ell + b]$ by the previous argument. The proof of part (a) is complete.

For part (b), we note that the variable term of the $\overline{\mathrm{MSE}}$ for $\ell \in [0, r - b]$ is

$$\frac{1}{b} \sum_{i=\ell+1}^{\ell+b} \frac{-a_i(\lambda)}{(\mu_i + \lambda)^2} = \frac{1}{b} \sum_{i=\ell+1}^{\ell+b} \frac{\lambda^2}{(\mu_i + \lambda)^2} - 1. \tag{24}$$

Since $i \mapsto \frac{\lambda^2}{(\mu_i + \lambda)^2}$ is an increasing function, summing it over a sliding window of length $b$ starting at $\ell + 1$, produces larger values as $\ell$ increases.

For part (c), the variable term of $\overline{\mathrm{MSE}}$ is again (24) for $b \in [1, r - \ell]$. The variable part is the average of the sequence $i \mapsto \frac{\lambda^2}{(\mu_i + \lambda)^2}$ over a window of length $b$. Increasing the window length then increases the average since the sequence is increasing. $\qquad\square$

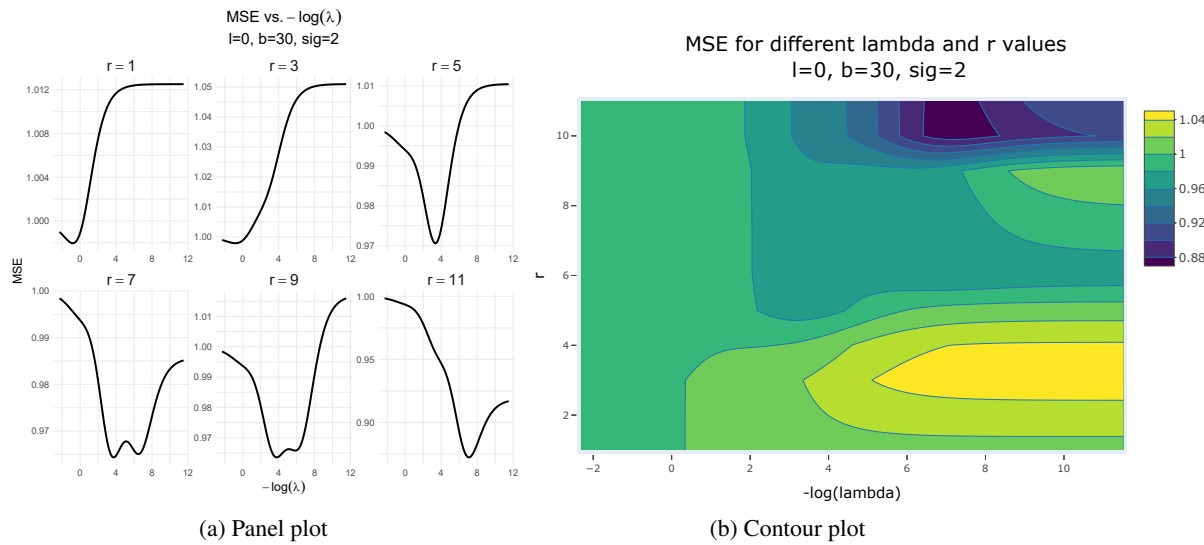

(a) Panel plot

(b) Contour plot

Figure 3: Multiple-descent and phase transition of $\lambda$-regularization curve: (a) Expected MSE as a function of $-\log(\lambda)$ for different values of $r$, and (b) overall contour plot of expected MSE for $r$ vs. $-\log(\lambda)$.

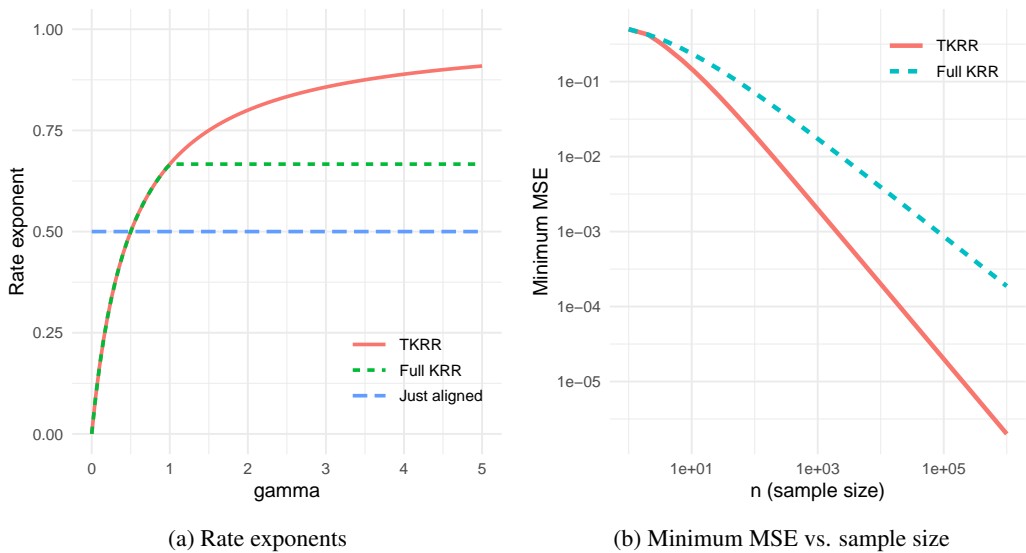

(a) Rate exponents

(b) Minimum MSE vs. sample size

Figure 4: (a) The rate exponent function $s(\gamma)$ for the TKRR, Eq. (17), compared with that of full KRR $s(\delta) = s(1 \wedge \gamma)$ and the minimax exponent over the RKHS ball $s(1/2)$. (b) The minimum achievable MSE by TKRR and full KRR, as a function of the sample size, when $\alpha = 1$ and $\gamma = 10$.

## C  Additional simulations

**Multiple-descent and phase transition**   We study the behavior of the $\lambda$-regularization curves for different values of the truncation parameter $r$ given a fixed noise level $\sigma$. The panel plot and corresponding contour plot are shown in Figure 3a and 3b, respectively. The plots show multiple-descent and phase transition as demonstrated in the $\lambda$-regularization curves for different values of the noise level and $r$-regularization curves shown in Section 5.

**Rate of TKRR vs. KRR**   We perform some experiments to coroborate the results of Theorem 2. We let the eigenvalues and TA scores decay polynomially with rates specified as in (13), and take the truncation parameter $r$ to be as derived in Theorem 2(a). For the full KRR and TKRR, we calculate the respective minimum value of MSE among 1000 values of the regularization parameter $\lambda$, evenly

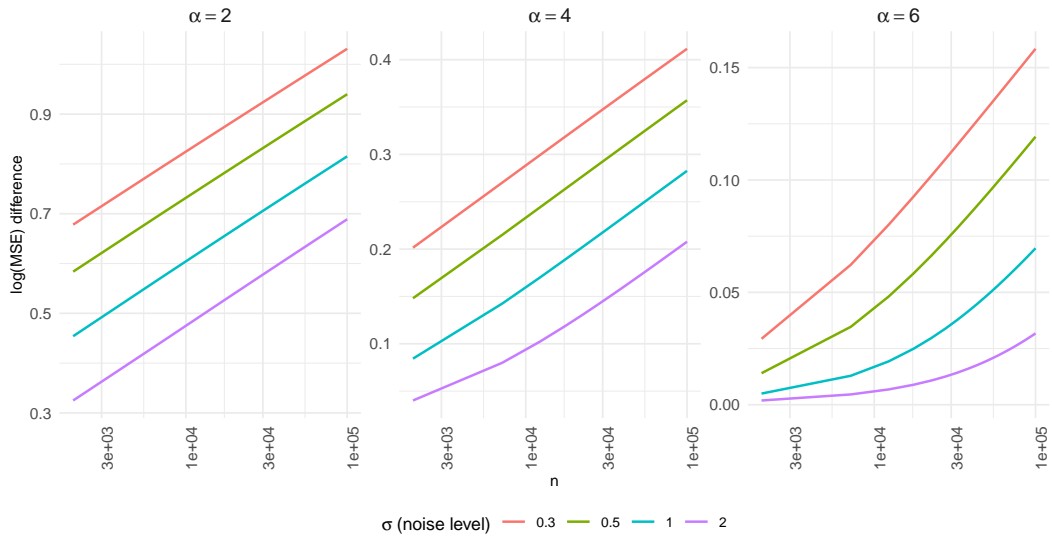

Figure 5: The difference of log(MSE) between the full KRR and TKRR versus the sample size $n$ for $\gamma = 5$, and various values of $\alpha$ and the noise level $\sigma$.

distributed between $10^{-10}$ and $10^2$. Figure 4(b) shows the MSE for the two methods, when $\alpha = 1$ and $\gamma = 10$, as a function of the sample size, on a log-log scale,. The difference in slope clearly shows the difference in rate between the two approaches.

The plots in Figure 5 show the difference in minimum log(MSE) between the full KRR and TKRR versus the sample size (on the log-scale), for different combinations of decay rate $\alpha$ and the noise level $\sigma$. For all the plots, we have $\gamma = 5$. According to Theorem 2, for sufficiently large $n$, the difference in minimum log(MSE) between the full KRR and TKRR should follow a line with positive slope when plotted as a function of $\log n$. This is clearly shown in Figure 5, where the positivity of the slope signifies the difference in rates between the two methods.

# D    RKHS background

Assume that $\mathcal{X}$ is a measurable space with a $\sigma$-finite measure $\mu$ and $\mathbb{H}$ is a separable RKHS over $\mathcal{X}$ with a measurable kernel $\mathbb{K} : \mathcal{X} \times \mathcal{X} \to \mathbb{R}$. We write $L^2 := L^2(\mu)$ for the $L^2$ space of functions from $\mathcal{X}$ to $\mathbb{R}$. For simplicity, we write $\|\mathbb{K}\|_{L^2}$ for the $L^2$ norm of the function $x \mapsto \sqrt{\mathbb{K}(x,x)}$. We assume

$$\|\mathbb{K}\|_{L^2} < \infty. \tag{25}$$

Then $\mathbb{H}$ is a subset of $L^2$ and the inclusion map $J : \mathbb{H} \to L^2$ is continuous. The adjoint of this map $J^* : L^2 \to \mathbb{H}$ is the following integral operator

$$J^* f(x) = \int \mathbb{K}(\cdot, x) f d\mu = \langle \mathbb{K}(\cdot, x), f \rangle_{L^2} \quad f \in L^2.$$

Let $T = JJ^* : L^2 \to L^2$. This can be thought of a the same integral operator acting on $L^2$ with output in $L^2$. The decomposition $T = JJ^*$ shows that $T$ is self-adjoint and positive. Condition (25) implies that $T$ is a Hilbert-Schmidt, and hence a compact, operator.

The spectral theorem for self-adjoint compact operators on $L^2$ implies that

$$Tf = \sum_{i \in I} \lambda_i e_i \langle f, e_i \rangle_{L^2} \quad \text{for all } f \in L^2$$

where $\{\lambda_i\}_{i \in I}$ are the non-zero eigenvalues of $T$ order in decreasing fashion and $\{e_i\}_{i \in I} \subset L^2$ a corresponding sequence of eigenvectors (at most countable), forming an orthonormal system (ONS) in $L^2$. That is, $Te_i = \lambda_i e_i$ and $\langle e_i, e_j \rangle_{L^2} = 1\{i = j\}$.

One can also view $\{e_i\}$ as functions in $\mathbb{H}$, and it is not hard to see that $\{e_i\}_{i \in I}$ is an orthogonal sequence in $\mathbb{H}$ with $\|e_i\|_{\mathbb{H}}^2 = 1/\lambda_i$. That is, $\langle e_i, e_j \rangle_{\mathbb{H}} = 1\{i = j\}/\lambda_i$. In other words, $\{\sqrt{\lambda_i} e_i\}_{i \in I}$ is an ONS in $\mathbb{H}$.

Assume from now on that we are dealing with a Mercer kernel $\mathbb{K}$, that is, $\mathcal{X}$ is a compact space and $\mathbb{K}$ is a continuous function. Then, we have the Mercer decomposition of the kernel function

$$K(x,y) = \sum_i \lambda_i e_i(x) e_i(y), \quad \forall x, y \in \mathcal{X} \tag{26}$$

where the convergence is uniform and absolute. It then follows that $\{\sqrt{\lambda_i} e_i\}$ is an orthonormal basis (ONB) of $\mathbb{H}$ and we have

$$\mathbb{H} = \Big\{ \sum_i \alpha_i e_i \mid \sum_i \frac{\alpha_i^2}{\lambda_i} < \infty \Big\}.$$

The treatment up to this point follows more or less the treatment in [23, Chapter 4].

From now on, we patch the sequence $\{e_i\}_{i \in I}$ to a complete orthonormal basis for the entire $L^2$ namely $\{e_i\}_{i \in I'}$ where $I'$ is a proper subset of $I$. Let $I_0 := I' \setminus I$. Then, $e_i, i \in I_0$ span the orthogonal complement of the image of $T$ (i.e., the null space of $T$). We let $\lambda_i = 0$ for $i \in I_0$, so that

$$Tf = \sum_{i \in I'} \lambda_i e_i \langle f, e_i \rangle_{L^2} \quad \text{for all } f \in L^2$$

still holds. The statement $\|e_i\|_{\mathbb{H}}^2 = \frac{1}{\lambda_i}$ also hold over $i \in I'$, interpreting $1/0$ as $\infty$. That is, $\|e_i\|_{\mathbb{H}} = \infty$ when $i \in I_0$, consistent with the fact that such $e_i$ are not in $\mathbb{H}$ (or more precisely do not have a version that is in $\mathbb{H}$).

## D.1 Target alignment

With this notation, every function in $L^2$ has a decomposition of the form $f = \sum_{i \in I'} \alpha_i e_i$ where $\alpha_i = \langle f, e_i \rangle_{L^2}$. Then, the RKHS $\mathbb{H}$ consists of those $f$ for which

$$\|f\|_{\mathbb{H}}^2 = \sum_{i \in I'} \frac{\alpha_i^2}{\lambda_i} < \infty. \tag{27}$$

One can think of either $\{\alpha_i\}_{i \in I}$ or $\{\alpha_i\}_{i \in I'}$ as the population level kernel alignment spectrum (that is, the population counterpart of Definition 1). Note that if $\alpha_i$ is nonzero for any $i \in I_0$, then $\|f\|_{\mathbb{H}} = \infty$ and that $f$ is not in $\mathbb{H}$. Even if $\alpha_i = 0$ for all $i \in I_0$, $\{\alpha_i\}_{i \in I}$ needs to decay as imposed in (27) for the function to belong to the RKHS. For example, a necessary condition is $\alpha_i = o(\sqrt{\lambda_i})$ for $i \in I_0$. In other words, belonging to the RKHS itself implies some amount of alignment between the target and the kernel (i.e. some level of decay for $\{\alpha_i\}$.)

To summarize, we can write

$$\mathbb{H} = \Big\{ f \in L^2 \mid \sum_{i \in I'} \frac{\langle f, e_i \rangle_{L^2}^2}{\lambda_i} < \infty \Big\}.$$

Let us connect to the setup of [9] and [12]. In short, these two papers impose the following condition

$$f = \sum_{i \in I'} \alpha_i e_i, \quad \sum_{i \in I'} \frac{\alpha_i^2}{\lambda_i^c} < \infty \tag{28}$$

for some $c \in [1, 2]$. If $c = 1$ this just means that $f \in \mathbb{H}$. If $c > 1$ it means that it is in a proper subset of $\mathbb{H}$. The $c$ here is the same as the $c$ in [9] and we have $c = 2r$ for parameter $r$ used in [12]. In our notation in this paper, $c = 2\gamma$. (Note that in our paper, $r$ is reserved to the spectral truncation level and is a different parameter.)

In addition [9] assumes $\lambda_i \asymp i^{-b}$ which is the same as the condition in [12], that is, $\lambda_i \asymp i^{-\alpha}$, for $\alpha = b$. Here, our notation matches that of [12]; see (13) which is the empirical counterpart of $\lambda_i \asymp i^{-\alpha}$. Also, [9] consider the case where $\lambda_i$ drop to zero exactly after some point (finite RKHS) which they refer to as the case $b = \infty$.

## D.2 Details of matching the setups

The conditions [9, 12] are not stated as cleanly as (28). Let us see how they can be reformulated in this equivalent fashion. The paper [9] which seems to be the origin of this condition works in the abstract setting of vector-valued RKHSs. We adapt the notation to the scalar-valued RKHSs. They work with operator $K_x : \mathbb{R} \to \mathbb{H}$ whose adjoint $K_x^* : \mathbb{H} \to \mathbb{R}$ is given by $K_x^* f = f(x)$ for every $f \in \mathbb{H}$. We then have $a K_x^* f = \langle K_x a, f \rangle_{\mathbb{H}}$ for any $a \in \mathbb{R}$ by the definition of an adjoint operator. Since $a K_x^* f = a f(x) = \langle a \mathbb{K}(\cdot, x), f \rangle_{\mathbb{H}}$, it follows that

$$K_x a = a \mathbb{K}(\cdot, x), \quad a \in \mathbb{R}.$$

Then, they define the operator $T_x := K_x K_x^* : \mathbb{H} \to \mathbb{H}$ and $T = \int_{\mathcal{X}} T_x d\mu(x)$. We have

$$\langle e_i, T_x e_j \rangle_{L^2} = \langle e_i, K_x K_x^* e_j \rangle_{L^2} = \langle e_i, K_x e_j(x) \rangle_{L^2}$$
$$= \langle e_i, e_j(x) K(\cdot, x) \rangle_{L^2}$$
$$= e_j(x) \langle e_i, K(\cdot, x) \rangle_{L^2} = e_j(x) \lambda_i e_i(x)$$

where the last step is since $e_i$ is an eigenvector of the integral operator $f \mapsto (x \mapsto \langle f, K(\cdot, x) \rangle_{L^2})$ from $L^2$ to $L^2$, and that the range of this operator is in fact in $\mathbb{H}$ (so evaluations make sense). This also follows from the Mercer decomposition. It then follows that

$$\langle e_i, T e_j \rangle_{L^2} = \int \langle e_i, T_x e_j \rangle_{L^2} d\mu(x) = \lambda_i \int e_i(x) e_j(x) d\mu(x) = \lambda_i \langle e_i, e_j \rangle_{L^2} = \lambda_i 1\{i = j\}.$$

That is $T$ can be viewed as a diagonal matrix $T = \mathrm{diag}(\lambda_i, i \in I')$ in the basis $\{e_i\}_{i \in I'}$.

The condition in [9] is $f = T^{(c-1)/2} g$ where $g \in \mathbb{H}$ (or more precisely $\|g\|_{\mathbb{H}}^2 \le R$). Let us write $g = \sum_{i \in I'} \beta_i e_i$ and $f = \sum_{i \in I'} \alpha_i e_i$. Since $T^{(c-1)/2}$ is a diagonal matrix in this basis, we have $\alpha_i = \lambda_i^{(c-1)/2} \beta_i$ or equivalently $\beta_i = \lambda_i^{(1-c)/2} \alpha_i$. Then, $g \in \mathbb{H}$ iff $\sum_i \beta_i^2 / \lambda_i < \infty$ which is equivalent to

$$\sum_i \frac{1}{\lambda_i} (\lambda_i^{(1-c)/2})^2 \alpha_i^2 < \infty \iff \sum_i \frac{\alpha_i^2}{\lambda_i^c} < \infty$$

and this is the desired condition.

Now to see that the condition in [12] is the same with $2r = c$, note that they require $\|\Sigma^{1/2-r} \theta^*\|_{\mathbb{H}} < \infty$ in their equation (7) which is a typo and is meant to be $\|\Sigma^{1/2-r} \theta^*\|_{\ell^2} < \infty$ in the $\ell^2$ sequence norm. Here $\Sigma = \mathrm{diag}(\lambda_i)$ in our notation.

As for $\theta^* = (\theta_i^*)$ which is a sequence in $\ell^2$, it is defined by the expansion $f^* = \sum_i \theta_i^* \psi_i$ where $\psi_i = \sqrt{\lambda_i} e_i$ in our notation. Thus, if we let $f = \sum_i \alpha_i e_i$, then $\alpha_i = \sqrt{\lambda_i} \theta_i^*$. So the condition imposed in [12] is

$$\sum_i (\lambda_i^{1/2-r} \theta_i^*)^2 < \infty \iff \sum_i (\lambda_i^{1/2-r} \lambda_i^{-1/2} \alpha_i)^2 < \infty \iff \sum_i \lambda_i^{-2r} \alpha_i^2 < \infty$$

which is the desired condition with $c = 2r$.

Immediately after stating this condition in [12], it is abandoned in favor of the condition $\lambda_i^{-2r} \alpha_i^2 \asymp i^{-1}$ which gives, together with $\lambda_i \asymp i^{-b}$

$$\alpha_i \asymp i^{-1/2} \lambda_i^r \asymp i^{-\frac{1+2rb}{2}}.$$

or equivalently $\theta_i^* \asymp \lambda_i^{-1/2} \alpha_i \asymp O(i^{-\frac{1+b(2r-1)}{2}})$. This is condition (8) in [12].

## D.3 Minimax rates

Theorem 1 and 2 in [9] together establish that the minimax rate for the signal model (28) when $c \in (1, 2]$ is given by $(1/\ell)^{bc/(bc+1)}$, where $\ell$ is the sample size. Moreover, the same rate is minimax for $c = 1$ up to logarithmic factors. Translating to our notation with $c = 2\gamma$, $\ell = n$ and $b = \alpha$, the minimax rate in our model is $(1/n)^{2\gamma\alpha/(2\gamma\alpha+1)}$ when $\gamma \in (1/2, 1]$, as claimed in Section 4.2.