# OpenReview forum: "Target alignment in truncated kernel ridge regression"
_NeurIPS.cc/2022/Conference — NeurIPS 2022 Accept_

### Official Review · Reviewer_Qioj · 2022-07-11

**Rating:** 5
**Confidence:** 2
**Soundness:** 3 good
**Presentation:** 2 fair
**Contribution:** 3 good

**Summary:**

this submission studies how the alignment between target function and the kernel in logistic kernel regression (KRR) can improve the regression fidelity. Kernel spectral truncation is used to adjust the alignment. The results indicate that: 1) for polynomial alignment, there is a regime that kernel truncation (TKRR) archives the parametric rate O(\sigma^2/n) which is faster than full KRR; 2) for bandlimited alignment, TKRR exhibits multiple descents.


**Questions:**

The connection to generalization is unclear. It’s mentioned in lines 59-61 that the results can be extended to generalization error. The setting however seems to be a pure inference task, with no training involved and no training/test data defined separately. This needs to be clarified.
The relation between \lambda and r is not clear. It seems that both of them are playing as regularization parameters and not clear if in certain regimes they overlap. A counter plot similar to Fig. 1 & 2 would be useful to clarify this issue for the readership.
It would be interesting to discuss connections to to singular value thresholding in the context of  (e.g., nuclear norm based regularization) which truncates certain singular value directions that are less aligned with the signal of interest.
In Fig. 2 (left) for noise intensity 0.18 and higher, it seems that the best choice for truncation is r=0. What does that mean?


**Limitations:**

They are properly discussed.


**Strengths And Weaknesses:**

Strength
theoretical novelty: this is the first work to study the impact of target alignment for KRR

Weakness
practical significance: it’s not clear how useful the findings about the alignment and concentration would be in practice. It heavily relies on the spectrum of f^* which is unknown, and no side information is known about it in this setting.
writing needs improvement: while the overall story is clear, this paper involves a lot of notations, where some details can be deferred to the appendix.

---

> ### Author Response · Authors · 2022-08-02
> **Response to Reviewer Qioj**
>
> > it’s not clear how useful the findings about the alignment and concentration would be in practice...
>
> Having theory is still useful in practice. For example, if you plot the error as a function of $r$, and see the non-monotonic behavior, you can guess that perhaps there are multiple disjoint bands in the alignment spectrum.
>
> That being said, there is a very good plugin estimator for the spectrum of $f\^\*$ in practice: Replacing $S\_x(f\^\*)$ with the noisy observations $y/\sqrt n$ in the definition of the spectrum gives a very good estimate, that is, $U\^T y / \sqrt n$ is a very good estimate of $U\^T S\_x(f\^\*)$. This is because each $u\_k\^T y / \sqrt n$ will effectively average over the noise in $y$. This can be made more precise using concentration inequalities and we plan to elaborate on that in the revision.
>
> > writing needs improvement: while the overall story is clear, this paper involves a lot of notations ...
>
> We have tried to move as much technical details as possible to the appendix. Please note that this is a theory paper. Without proper notation, it will be hard for people to follow. In the revision, we will try to discuss the results more in plain English, so hopefully that will help.  Please also note that two other reviewers found the paper, to quote "well-written and the ideas and the results are presented clearly," and "easy to follow and well-written."
>
> > The connection to generalization is unclear...
>
> Thanks for your comment. We will make the connection more clear in the revision. Here is a brief summary: The generalization error really makes sense in the random design setting. Let $(x,y)$ be a random test point, and let $(x\_1,y\_1),\dots,(x\_n,y\_n)$ be i.i.d. training data, all from the same joint distribution $\mathbb P$ on $(x,y)$. Let $\mathbb P\_X$ be the marginal distribution of $x$ under $\mathbb P$. The generalization error for a fixed function $f$ is
>
> $\mathbb E(y - f(x))\^2 = \mathbb E (f\^\*(x) - f(x) + w)\^2 = \mathbb E(f\^\*(x) - f(x))\^2 +\sigma\^2$,
>
> where the expectation is taken w.r.t. the randomness in both $x$ and $y$. This can further be written as $\\| f - f\^\* \\|\_{\mathbb P\_X}\^2 + \sigma^2$, that is, the population $L\^2$ norm-squared of $f - f\^\*$ plus the variance of the noise. The variance of the noise is the unimporvable part of the generalization error, i.e., the minimum Bayes risk. So the excess generalization error is $\\| f - f\^\* \\|\_{\mathbb P\_X}\^2$.  For large $n$, since the $L\^2$ norm is an integral, this can be well-approximated by the empirical norm based on the training $x\_i$ data, that is, $\frac1n\sum\_{i=1}\^n (f(x\_i) - f\^\*(x\_i))^2$ which is the empirical norm that we have considered in the paper. This is why we call it the empirical excess generalization error in line 61. This approximation can also be made more precise; we have elaborated on this in response to Reviewer UAiV and plan to include those details in the revision.
>
> > The relation between $\lambda$ and r is not clear. It seems that both of them are playing as regularization parameters and not clear if in certain regimes they overlap. A counter plot similar to Fig. 1 \& 2 would be useful to clarify this issue for the readership.
>
> Yes, the relation is in general complicated. Our Theorem 2 shows that in the case of polynomial alignment, one needs both to achieve the best performance.
> Thanks for the suggestion about the $r$-$\lambda$ contour plot; that is a great way to show the complicated nature of their joint effect on the performance. We have made the plot and will add it to the revision.
>
> > It would be interesting to discuss connections to to singular value thresholding ...
>
> Thanks for the suggestion. We can make the following connection: Our results show that spectral truncation reduces the variance, the third term in Eqn. (9), and this is in line with what singular value thresholding does by reducing the noisy directions. We will add a few sentences about this to the “Conclusion” section.
>
> > In Fig. 2 (left) for noise intensity 0.18 and higher, it seems that the best choice for truncation is r=0. What does that mean?
>
> Thanks for the careful observation. The plot is a bit misleading. The minimum truncation level is $r = 1$, and that is what those plot should show. We will make the x-axis range on these plots more clear. What the plots show is that for very large noise levels, the best performance is achieved if we truncate right away, that is, only keep the first component from the alignment spectrum. This is in line with our theory developed in Proposition 2(a), although perhaps not clearly discussed in the present manuscript. Your comment here is very much related to that of Reviewer UAiV on the relation between the optimal truncation level and $\sigma^2$. We plan to make this much more clear in the revision, as elaborated in response to their comment.

---

> > ### Comment · Reviewer_Qioj · 2022-08-08
> > **questions clarified**
> >
> > I would like to thank the authors for providing the response. I found the discussion useful to clarify my questions to understand the paper. I would keep my score. Please add clarifying statements along those comments in the revised paper to make it more readable.

---

> > > ### Author Response · Authors · 2022-08-09
> > > **Thank you**
> > >
> > > Thank you! We are happy that our comments clarified your questions. Yes, we will add the clarifying statements to the revised paper.

---

### Official Review · Reviewer_UAiV · 2022-07-11

**Rating:** 6
**Confidence:** 3
**Soundness:** 3 good
**Presentation:** 2 fair
**Contribution:** 2 fair

**Summary:**

The manuscript provides an analysis of truncated kernel ridge regression. In particular, the authors show how alignment of the target function influences convergence rates, and identify scenarios in which TKRR exhibits stronger guarantees than ridge regression. In addition, they show examples in which generalisation performance of TKRR demonstrates non-monotonicity and phase transition behaviour in regularisation and truncation parameters.

**Questions:**

Please, see weaknesses section above and correct/comments on any misunderstandings from my side

**Limitations:**

Yes, they have

**Strengths And Weaknesses:**

Strengths:

The main result of the paper is the exact expression for mean squared error of TKRR from which the authors derive two interesting conclusions:
1. They demonstrate how a non-monotonous behaviour of the generalisation performance with respect to truncation parameter takes place, which provides a new insight into double-descent phenomena
2. They derive stronger convergence results for TKRR (compared to canonical KRR) in case of strong alignment between the target and kernel functions, which are new to the best of my knowledge

Weaknesses:

1. All analysis in the paper is performed in the fixed design setting. In the conclusion the authors state that it can be extended to the random design model. However it’s outside of my area of expertise, so I cannot judge how straightforward/possible that extension is, taking into account also that TKRR itself is data-dependent
2. Theoretical results presented in the paper indicate importance of the noise level for selection of a suitable truncation parameter. This is also supported by simulation results, where figure 2a demonstrates that whether choosing a larger r (already being in decreasing error regime) is a beneficial approach depends on the noise level. With this in mind, I believe the manuscript would benefit from a more thorough discussion of this aspect of the problem, in particular what can be deducted from the theoretical results and what only from the simulations/experiments
3. I found reading of the manuscript rather strenuous. While in parts it can be attributed to its level of technicality, I would suggest the authors to consider some re-organisation of the material. In particular, I would recommend at least partially incorporating discussion currently presented in Appendix D2 and D3 into the main text, potentially at the expense of the proofs in Section 7.
4. In line 240, I believe the reference should be to Proposition 2a instead of Theorem 2.

---

> ### Author Response · Authors · 2022-08-02
> **Response to Reviewer UAiV**
>
> > 1. All analysis in the paper is performed in the fixed design setting. In the conclusion the authors state that it can be extended to the random design model ... I cannot judge how straightforward/possible that extension is ... .
>
> Thanks. Our original intention was not to provide a rigorous statement in the random design case, and there could be obstacles that we did not anticipate. However, we will add a sketch of what we believe is possible to the supplementary material. Basically it boils down to this: Assume that the RKHS is generated by a Mercer kernel (continuous function on a compact space). Then, the RKHS is a convex class of uniformly bounded functions. Then, Theorem 14.12 in [26] applies and we have $\frac12 \\|f\\|\_{L^2}^2 \le  \\|f\\|\_n^2$ uniformly over $\mathbb H$, with probability $1 - c\_1 e^{-c\_2 n \delta\_n^2}$ where $\delta_n$ is a critical radius that goes to zero with $n \delta\_n^2 \to \infty$. We can then apply this result to get $\\|\widehat f\_{r,\lambda} - f\^\*\\|\_{L^2}^2 \le 2 \\|\widehat f\_{r,\lambda} - f\^\*\\|\_{n}^2$ with high probability. The LHS of this inequality is the excess generalization error and the RHS is what we control. Combined with Markov inequality, Theorem 1 then shows that the same data-dependent bound on the LHS of (10) is also a high probability bound on the generalization error in the random design setting.
>
> The only caveat in the above argument is the zero-mean assumption on the function class in Theorem 14.12 of [26]. We do not think it is needed and we would like to avoid it. If we manage to prove a result, we will make a rigorous statement in the revision. Otherwise, we will point out the sketch above and leave the details to future work.
>
> > Theoretical results presented in the paper indicate importance of the noise level for selection of a suitable truncation parameter...
>
> This is a great suggestion! There is already an indication of how the noise level plays a role in deciding the optimal truncation level, in the definition of $j\^\*$ in Proposition 2(a). For sufficiently small levels of the noise, $j\^\*$ will be equal to $\ell+1$ (as discussed on line 169). Then, for $r < \ell+1$, the MSE is increasing, but the moment we enter the signal band ($r = \ell+1$), the MSE starts to decrease and keeps decreasing till we get to the end of the band ($r = \ell+b)$, at which point it starts to increase again. So, in this case (small enough noise level), the optimal truncation level is at the end of the band, i.e., $r = \ell+b$. If we increase the noise level, $j\^\*$ will be in the middle of the band. In this case, depending on where in the band $j\^\*$ lies, the optimal truncation level could still be $r = \ell+b$. But at some point, the noise level is so high that $j\^\*$ is very close to the end of the band, in which case, the dip in the MSE over the band is so small that the best truncation is just at $r=1$ level. This includes the case where there is no $j\^\*$ in the band, hence the MSE is monotonically increasing (very high noise levels).
>
> The situation in Figure 2a is a bit more complicated since we have two bands, and formally we have not stated a result for this case. But a similar pattern to the above holds. We will try to state a formal result in the two-banded case in the revision if possible, and map out the implications. In any case, hopefully, the discussion above is enough to address your point, and we plan to add it to the revision.
>
> > I found reading of the manuscript rather strenuous ... partially incorporating discussion currently presented in Appendix D2 and D3 into the main text ...
>
> We debated your suggestion but we prefer to keep the current format. The discussion in Appendix D2 and D3 is a fairly  technical translation of the results of other papers to our notation. It is more of an expository note on existing papers and including them in the main text will detract attention  from our own contributions. We believe the main message is adequately carried in the main text in the current format. Perhaps with additional clarifications that we will add in the revision in response to reviewers, it will become easier to read. Also, please note that two other reviewers found the paper, to quote "well-written and the ideas and the results are presented clearly," and "easy to follow and well-written."
>
> > In line 240, I believe the reference should be to Proposition 2a instead of Theorem 2.
>
> Thanks! We will fix this in the revision.

---

> > ### Comment · Reviewer_UAiV · 2022-08-06
> > **Thank you for your reply**
> >
> > Dear authors,
> >
> > thank you for your reply, it clarified my confusions and if you include some of these points in the manuscript, I believe it will help the readers as well.
> >
> > Just one last comment on the random design discussion - it of course would be great to have a formal result. However, if that's not possible, I would ask you to re-phrase the conclusion statement from "... it is possible to extend ..." to something softer, as otherwise it feels a bit awkward that something that seems surely possible to do is not in fact done.
> >
> > Otherwise, I'm happy to keep my positive rating of your submission.

---

> > > ### Author Response · Authors · 2022-08-09
> > > **Thank you**
> > >
> > > > thank you for your reply, it clarified my confusions and if you include some of these points in the manuscript, I believe it will help the readers as well.
> > >
> > > We are glad that it cleared the issue. We will include these points in the revised manuscript.
> > >
> > > > ... if that's not possible, I would ask you to re-phrase the conclusion statement from "... it is possible to extend ..." to something softer, as otherwise it feels a bit awkward that something that seems surely possible to do is not in fact done.
> > >
> > > Yes, this is very reasonable. We will follow your advice and tone down the claim if we did not manage to state a formal statement.
> > >
> > > > Otherwise, I'm happy to keep my positive rating of your submission.
> > >
> > > Thank you!

---

### Official Review · Reviewer_6bFj · 2022-07-11

**Rating:** 6
**Confidence:** 4
**Soundness:** 3 good
**Presentation:** 3 good
**Contribution:** 3 good

**Summary:**

The paper considers kernel ridge regression (KRR)  and its truncated version for estimating a target function contaminated with the i.i.d. noise. Two situations are considered depending on the target alignment (TA) scores, which are the coefficients of projection of the underlying target vector on the eigenvalues of normalized empirical kernel matrix K. 1)TA scores are non-zero in a few consecutive components, in which they sampled as i.i.d. with non-zero mean, 2) TA scores are decaying polynomially as well as eigenvalues of $K$. In the first example, the authors demonstrate possible non-monotonous behavior of the MSE depending on truncation parameter r, and other parameters fixed. In the second one, four performance regimes are considered for truncated and full KRR based on the strength of alignment of the target function with the eigenvectors of $K$. If the target alignment coefficients decay fast, then with the proper truncation and smoothing parameter the MSE achieves a better rate than the full KRR with the parametric rate as the limit in the decay parameter.


**Questions:**

Is the statement about the solution of (6) being not unique correct?
(6) is a ridge regression problem with a design matrix with $r$ orthogonal columns, $r<n$  (see line 128)?
In the supplementary material (17) has a non-unique solution, at the same time
$$\tilde{S}^*_{\mathbf{x}}(w)(x) = \frac{1}{\sqrt{n}}\sum_{j=1}^n w_j \tilde{\mathbb{K}}(x,x_i) = \frac{1}{\sqrt{n}} \sum_{k=1}^r \mu_k \phi_k(x) \sum_{j=1}^n \phi_k(x_j)  w_j  =  \sum_{k=1}^r \mu_k \phi_k(x)   u_k^{\top} w.$$
 Multiple solutions in terms of $w$  in (17) come from the fact that $w$ contains an arbitrary vector, that lies in the subspace orthogonal to the one spanning $\{u_1,\dots,u_r\}$, but taking $u_k^{\top} w$ leaves only one solutions?

It would be worth commenting the results of the submission in comparison with Tsigler, A., & Bartlett, P. L. (2020). Benign overfitting in ridge regression. arXiv preprint arXiv:2009.14286.

----------------
I increased the rating after the authors commented on the questions.

**Limitations:**

The limitations and well addressed in the work. No negative societal impact is anticipated.

**Strengths And Weaknesses:**

Strengths:
The paper is easy to follow and well written. The examples considered serve as a good illustration of the double-descent phenomenon in RKHS and the dependence of generalization performance of the (T)KRR  on properties of RKHS and on how well the underlying function is aligned with the eigenvectors of $K$.

Weaknesses:
-  I am not sure I understand section 3.1, see the questions below.
- The evaluation is illustrative enough, but only uses the artificial dataset.

Minor details:
- w is used for both noise  in (1) and coefficient vector in (3)
- is $\sqrt{N}$ needed in the formula between the lines 117-118?
- misprint:  $-1$ in the power of $\sigma/n$ in 192

---

> ### Author Response · Authors · 2022-08-02
> **Response to Reviewer 6bFj**
>
> Thanks for your careful reading! We will address the uniqueness issue below,  but please note that the uniqueness does not affect the main results of the paper. The material in Section 3.1 is more of an interesting side note and somewhat tangential to the central point of the paper which is discussed in Section 4. We couldn't find any criticism of the main points of the paper in your review and hence are somewhat puzzled by your rating.
> We would be happy to address any shortcomings you find regarding the main message of the paper.
>
> > I am not sure I understand section 3.1, see the questions below...  Is the statement about the solution of (6) being not unique correct? ...
>
> Your point is valid and the solution is indeed unique. Thanks for pointing this out! There is a subtelty in how one defines the functional version of the TKRR estimate which perhaps led us not to notice this. The common way of defining the TKRR estimate is to pass $\widetilde \omega$ through the adjoint operator associated with the original RKHS, that is to form $S\_x\^\*(\widetilde \omega)$ where $\widetilde \omega$ is the solution of (17) in the supplementary material. This form will inherit non-uniqness of $\widetilde \omega$. The way we stated Prop.~1 is more approperiate for this form.
>
> The way we defined the TKRR estimate in (6) though leads to passing $\widetilde \omega$ through the adjoint associated with the smaller RKHS, that is $\widetilde S\_x\^\*(\widetilde \omega)$, which as you point out will be the same for all the solutions of (17). This is even better than we anticipated! It reinforces the idea that (6) is the canonical way of defining the TKRR functional form. Proposition 1 as stated is still technically correct in this case (the solution set in part (b) will just be a singleton), but can be simplified greatly given the uniqueness.
>
> We plan to revise Prop. 1 and point out the uniqueness. On the other hand, no matter which of the two versions of the functional estimate of TKRR one wants to use, the fact that they all map to the same point under the sampling operator $S_x$, remains true. We intend to point this out and perhaps elaborate in the supplement. This makes the subsequent results valid for either version.
>
> > The evaluation is illustrative enough, but only uses the artificial dataset.
>
> Since our paper is a theoretical one, we believe simulations are more suited for demonstrating the results. Real datasets have many moving parts which are hard to control, hence not that effective in demonstrating our desired points. It is a common practice in theoretical work to either not include any experiments at all (like the Tsigler \& Bartlett paper below) or to provide experiments in simulated settings.
>
>
> > It would be worth commenting the results of the submission in comparison with Tsigler, A., \& Bartlett, P. L. (2020).
>
> Thanks for the pointer. We will discuss the connections in the revision. There are interesting parallels but also notable differences between our work and theirs. They consider the usual ridge regression while we work with the kernel ridge, although the results can be translated back and forth after some transformation.
>
> Interestingly, they also have a spectral truncation parameter $k$, although in Tsigler \& Bartlett, the truncation level is more of a device in the proof that can be optimized to obtain the tightest upper bound, whereas the truncation level $r$ in our case is a regularization parameter in TKRR that can be tuned in practice. So the bound in Tsigler \& Bartlett will essentially correspond that of the Full KRR in our paper. In fact, we also use a similar proof-device $k$ in the proof of Theorem 2, separate from the $r$-truncation level of TKRR (see line 272 for the definition of $k$). As an important corollary, our results about TKRR cannot be deduced from those in Tsigler \& Bartlett.
>
> Another point of difference is the focus in Tsigler \& Bartlett which is on how the eigendecay of the kernel (or equivalently the covariance matrix) affects the MSE bound, while we focus on how the interaction of the target alignment decay and the kernel eigendecay together affect the bounds. There are indications of the effect of the target alignment in Tsigler \& Bartlett, in the form of the tail energy of their $\theta\^\*$ parameter, but the implications of this decay seems to have not been explored in detail compared to our work.
>
> > w is used for both noise in (1) and coefficient vector in (3)
>
> They are different symbols. The one in (1) is $w$ and the one in (3) is omega $\omega$. We will change the noise vector in (1) to $\varepsilon$ to avoid confusion.
>
> > is $\sqrt N$  needed in the formula between the lines 117-118?
>
> Thanks for pointing this out. You are correct that it is not needed.  It will be fixed in the revision. (This doesn't affect the subsequent developments).
>
> > misprint: -1 in the power of  in 192
>
> Thanks again! We will fix it.

---

> > ### Comment · Reviewer_6bFj · 2022-08-08
> > **reply**
> >
> > Dear authors,
> >
> > Thank you for your response! Assuming the corrections would be incorporated in the revised paper, I will change the rating to a positive one.

---

> > > ### Author Response · Authors · 2022-08-09
> > > **Thank you**
> > >
> > > Thank you for your positive vote! Yes, we will incorporate the corrections in the revised paper.

---

### Official Review · Reviewer_x1n8 · 2022-07-17

**Rating:** 8
**Confidence:** 5
**Soundness:** 4 excellent
**Presentation:** 4 excellent
**Contribution:** 4 excellent

**Summary:**

The paper studies a truncated KRR method and shows how the alignment of the target function with the kernel improves the performance of the TKRR. The paper also sheds light on a situation when the TKRR can achieve parametric rates of convergence - something which is not possible for KRR. Both a theoretical analysis and a simulation study is included in the paper.

**Questions:**

None.

**Limitations:**

No potential negative societal impact.

**Strengths And Weaknesses:**

The paper is sufficiently novel and studies a very interesting variant of KRR, namely, TKRR and demonstrates both theoretically and empirically the strength of TKRR, in particular, when the target function is aligned with the kernel function. The paper is well-written and the ideas and the results are presented clearly.

---

> ### Author Response · Authors · 2022-08-02
> **Response to Reviewer x1n8**
>
> Thank you for your positive feedback and encouragement!

---

### Author Response · Authors · 2022-08-02
**General comments to the reviewers**

We thank all the reviewers for their valuable comments which has helped us improve the presentation of the paper. We address specific comments in response to reviewers individually. A summary of the main changes that we plan in the revision are as follows:

- We will address and correct the uniqueness issue raised by Reviewer 6bFj.

- We will add a discussion of connections with the work of Tsigler \& Bartlett (2020) as detailed in response to Reviewer 6bFj.

- We will elaborate on how the results can be extended to the random design (response to Reviewer UAiV) and to the generalization error (response to Reviewer Qioj). The two issues are related.

- We will discuss how target alignment spectrum can be estimated in practice (response to Reviewer Qioj).

- We will clarify the role of the noise level in choosing the optimal truncation level (response to Reviewer UAiV).

- We will include plots showing the interaction of $\lambda$ and $r$ (response to Reviewer Qioj).

We would be happy to address any outstanding concerns that you might have.

---

### Meta-Review · Area_Chair_VAGM · 2022-08-26

**Recommendation:** Accept
**Confidence:** Certain

**Metareview:**

The reviewers all found that the paper is interesting and contributes interesting new results. The reviewers have made several useful constructive comments that the authors are strongly encouraged to take into account.

**Award:**

No

---

### Decision · Program_Chairs · 2022-09-14

Accept